# HYPER-SET: DESIGNING TRANSFORMERS VIA HYPERSPHERICAL ENERGY MINIMIZATION

**Yunzhe Hu**[1]**, Difan Zou**[1,2]**, Dong Xu**[1]
[1]School of Computing and Data Science, The University of Hong Kong
[2]Shenzhen Loop Area Institute
`{yzhu,dzou,dongxu}@cs.hku.hk`

## ABSTRACT

Transformer-based models have achieved remarkable success, but their core components, Transformer layers, are largely heuristics-driven and engineered from the bottom up, calling for a prototypical model with high interpretability and practical competence. To this end, we conceptualize a principled, top-down approach grounded in energy-based interpretation. Specifically, we formalize token dynamics as a joint maximum likelihood estimation on the hypersphere, featuring two properties: semantic alignment in the high-dimensional space and distributional uniformity in the low-dimensional space. By quantifying them with extended Hopfield energy functions, we instantiate this idea as a constrained energy minimization problem, which enables designs of symmetric attention and feedforward modules with RMS normalization. We further present *Hyper-Spherical Energy Transformer* (HYPER-SET), a recurrent-depth alternative to vanilla Transformers naturally emerging from iterative energy optimization on the hypersphere. With shared parameters across layers, HYPER-SET can scale to arbitrary depth with fewer parameters. Theoretically grounded and compact, it achieves competitive or superior performance across diverse tasks, including Sudoku solving, image classification, and masked image modeling. We also design novel variations under the proposed general principle, such as linear attention and gated feedforward layer, and showcase the scalability with depth-wise LoRA. Our results highlight HYPER-SET as a step toward interpretable and principled Transformer design. Code is available at https://github.com/huyunzhe/hyper-set.

## 1 INTRODUCTION

Transformer-based models (Vaswani et al., 2017) have become foundational across diverse domains, including computer vision (Dosovitskiy et al., 2021; Bao et al., 2022; He et al., 2022; Peebles & Xie, 2023), natural language (Devlin et al., 2019; Lan et al., 2020; Brown et al., 2020), robotics (Brohan et al., 2022; Chen et al., 2021), and scientific discovery (Jumper et al., 2021; Kamienny et al., 2022). In recent years, there has been evidence that scaling up model size, dataset size, or computational budget during pre-training can yield unprecedented performance gains (Kaplan et al., 2020), driving the proliferation of Transformer-based foundation models (OpenAI et al., 2024; Dubey et al., 2024; Anil et al., 2023; Oquab et al., 2024).

Despite these achievements, the architecture of Transformers—especially the role of individual layers—remains largely heuristic. For instance, empirical studies have observed high redundancy in the deeper layers (Gromov et al., 2025; Men et al., 2025), uniformity of representations in the middle layers (Sun et al., 2025), and robustness to permuting certain intermediate layers (Lad et al., 2025) in LLMs. These findings suggest convergent functionality that one layer represents, yet our understanding of its role in processing information and representation learning remains limited. While interpretability efforts to unveil the function underlying network layers, especially Transformer blocks, exist—spanning mechanistic interpretability (Elhage et al., 2021; Nanda et al., 2023; Wang et al., 2023; Huben et al., 2024), causal mediation analysis (Vig et al., 2020; Meng et al., 2022), and visualization (Bricken et al., 2023; Olsson et al., 2022)—most of them focus on *post-hoc* interpretation and phenomenological approaches. This motivates a pivotal question:

> *Can we find or design a function prior that induces a model interpretable by construction?*

One approach to achieving intrinsic interpretability is to embed an explicit optimization process into neural networks, known as model-based deep learning (Shlezinger et al., 2023). Prior works have designed networks that solve domain-specific problems such as constraint satisfaction (Wang et al., 2019), optimal control (Amos & Kolter, 2017; Amos et al., 2018), or physical simulation (Greydanus et al., 2019; Karniadakis et al., 2021). However, these models often rely on fixed task priors and lack generality.

Another more general avenue is energy-based learning (EBL) (Dawid & LeCun, 2024), which frames prediction as minimizing a scalar energy function $E_\theta(x, y)$ over outputs $y$ conditioned on inputs $x$. Within this framework, Energy Transformer (Hoover et al., 2024) interprets Transformer layers as iterative optimization over the canonical continuous Hopfield energy (Ramsauer et al., 2021; Krotov & Hopfield, 2021) yet focuses on mechanistic analogies to associative memory without grounding its formulation in specific representational challenges. In contrast, our goal is to find a design principle from top down that can not only reinterpret existing components but is also generalizable to novel architecture design *constructively*.

In this work, we therefore take a fundamentally different approach by introducing a principle grounded in maximum likelihood estimation (MLE) for tokens on the hypersphere. Under mild assumptions, we interpret it under two complementary objectives for representation dynamics: semantic alignment (mode seeking) in high-dimensional space and distributional uniformity (mass covering) in a low-dimensional subspace. To translate these objectives into optimizable quantities over tokens, we define two complementary Hopfield-style energy functions that quantify these objectives and can be minimized through iterative optimization. This leads to the *Hyper-Spherical Energy Transformer* (HYPER-SET)—a recurrent-depth model in which core components such as symmetric attention, feedforward layers, RMSNorm, and skip connection emerge naturally from the optimization dynamics. With only one set of shared parameters across iterations, HYPER-SET is compact, interpretable by design, and empirically competitive across diverse tasks, including reasoning, classification, and masked image modeling. Beyond a single instantiation, this principle can induce novel architectural designs by generalizing the energy functions, enabling variants such as linear attention and gated feedforward layer. To enhance scalability, we introduce depth-wise low-rank adaptation (LoRA), allowing flexible iteration-specific modulation with minimal parameter overhead. Our key contributions are summarized as follows:

1. **Theoretical Formulation**: We conceptualize a general principle for information processing in layer dynamics based on maximum likelihood estimation on the hypersphere with two properties: uniformity and alignment, quantified via complementary Hopfield-style energy functions.

2. **Energy-Driven Architecture**: We derive a compact Transformer-based model through pure energy minimization, where core components—including symmetric attention, feedforward, RMSNorm (Zhang & Sennrich, 2019), and skip connection—emerge naturally.

3. **Competitive Performance**: We show competitive performance to vanilla Transformer across reasoning, classification, and masked modeling while demonstrating generality to design novel components (e.g., linear attention, gated feedforward) and scalability with flexible computation.

## 2 RELATED WORK

### 2.1 ENERGY-BASED LEARNING

Energy-based learning (EBL) (LeCun et al., 2006; Dawid & LeCun, 2024) provides a unifying framework for modeling prediction as minimizing an energy function. Early forms include Hopfield networks (Hopfield, 1982) and Boltzmann machines (Ackley et al., 1985). Modern developments in EBL span both generative modeling—via energy functions (Du & Mordatch, 2019) or their gradients (as in score-based models (Sohl-Dickstein et al., 2015; Song & Ermon, 2019))—and representation learning. Another line of work views network layers as the result of iterative energy minimization. Some approaches define energy implicitly through neural networks (Bai et al., 2019; Du et al., 2022; 2024), while recent work Energy Transformer (Hoover et al., 2024) draws analogies between

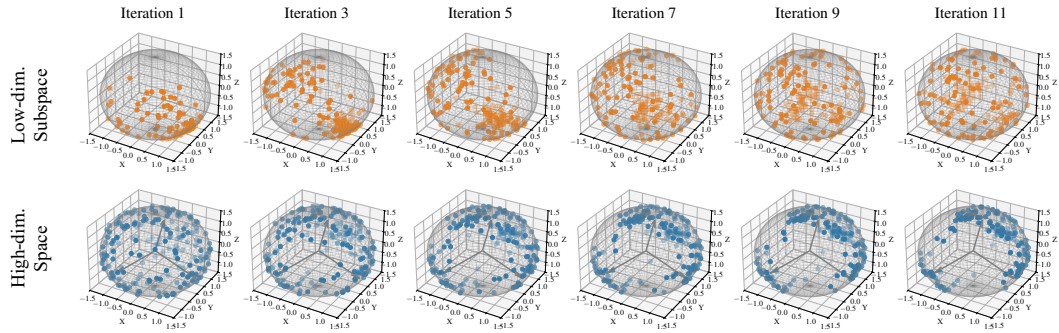

Figure 1: Evolution of tokens in the forward pass. *Top*: Tokens projected onto subspaces are progressively separated on the **low-dimensional** hypersphere. *Bottom*: Tokens gradually align with anchor vectors in the **high-dimensional** hypersphere. Visualization is carried out in three-dimensional space for illustrative purposes.

attention layers and explicit energy descent but mainly focuses on reinterpretation rather than principled derivation. Our work differs in that we design the Transformer block by quantifying a general principle that can induce variants through alternative energy.

Other works also explore energy formulations on the hypersphere (Liu et al., 2018; Loshchilov et al., 2025), but mostly in the weight space. By contrast, we define our energies directly on the representation space. Additionally, recent theoretical studies on memory capacity in modern Hopfield networks (Hu et al., 2024a; Wu et al., 2024a; Hu et al., 2024b) emphasize spreading patterns on the sphere but focus primarily on memory retrieval and cross-attention.

## 2.2 MODEL DESIGN FROM FIRST PRINCIPLES

While neural network architectures are often shaped by engineering practices, recent work has explored designing or interpreting them through principled lenses like signal processing, information theory, and neurobiology. For example, deep unrolling of the sparse coding algorithms has led to the development of fully connected networks (Gregor & LeCun, 2010), convolution networks (Papyan et al., 2017; 2018), and even graph neural networks through iterative algorithms (Yang et al., 2021). Similarly, the sparse rate reduction principle has been used to derive the Transformer architecture (Yu et al., 2023). Other approaches draw inspiration from approximation theory (Liu et al., 2025) and brain computation (Kozachkov et al., 2023), further bridging the gap between theoretical insights and practical network design.

## 2.3 DEPTH RECURRENCE IN TRANSFORMERS

Recurrence has been scientifically considered as a core computational mechanism in the biological visual system enabling flexible computational depth, integration of priors, and efficient allocation of compute under resource constraints (van Bergen & Kriegeskorte, 2020). Depth-wise recurrence in Transformer architectures, under the name of universal (Devlin et al., 2019), recursive/recurrent (Bae et al., 2025; Geiping et al., 2025) or looped Transformers (Giannou et al., 2023), where cross-layer weights are shared, has emerged as a critical avenue for reducing parameters and enabling iterative reasoning. Its repeated reuse of the same layer has been demonstrated to be capable of emulating iterative algorithms (Schwarzschild et al., 2021; Saunshi et al., 2025). Crucially, it has been demonstrated with particular strength mostly on systematic generalization (Csordás et al., 2021) and structured reasoning tasks (Schwarzschild et al., 2021; Bansal et al., 2022), while its applications on general-domain tasks that need hierarchical processing (e.g., image perception) are under-explored.

## 3 CONCEPTUALIZATION AND INSTANTIATION

To answer the introductory question, we conjecture that effective representations should exhibit two complementary properties: **semantic alignment** in a high-dimensional space and **distributional uniformity** in a low-dimensional subspace. This dual perspective reflects the balance of *mode seek-*

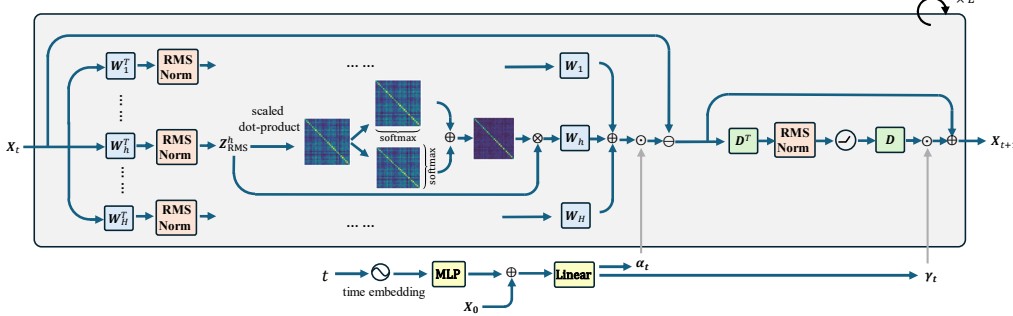

Figure 2: Overview of hyperspherical energy Transformer layer. It recovers sequential stacking of symmetric self-attention, feedforward, skip connection, and RMSNorm from sheer minimization of extended Hopfield energy. Adaptive step sizes are learned given the current $t$ and initial input $\boldsymbol{X}_0$.

*ing* and *mass covering*—terms we use to characterize the interplay between information preservation and entropy collapse prevention in representation learning. Figure 1 shows an illustrative example.

We formalize this conceptualization under maximum likelihood estimation. Specifically, we instantiate the forward dynamics as an optimization over a token-level vector $\boldsymbol{x}$ balancing two terms:

$$\min_{\boldsymbol{x}} \sum_{h=1}^{H} \underbrace{D_{\mathrm{KL}}\left(p(\boldsymbol{z})\|p_\phi(\boldsymbol{z}^h \mid \boldsymbol{x})\right)}_{\text{uniformity}} \underbrace{- \log p_\theta(\boldsymbol{x})}_{\text{alignment}}, \tag{1}$$

where $\boldsymbol{z}^h$ represents low-dimensional projections of the high-dimensional representation $\boldsymbol{x}$.

The first term encourages the projections $\boldsymbol{z}^h$ to approximate a prior uniform distribution $p(\boldsymbol{z})$ on a hypersphere, thus maximizing entropy and mitigating representational collapse. The second term promotes alignment between $\boldsymbol{x}$ and mean directions, which can be modeled using von Mises–Fisher distributions. A detailed justification and interpretation of this objective is provided in Appendix A.

This objective resonates with but differs from the contrastive learning objective that unifies alignment and uniformity in a shared latent space (Wang & Isola, 2020). Our work instead takes on an energy view to quantify these two key ingredients into optimizable functions of $\boldsymbol{x}$ that can induce Transformer architectures.

## 4    HYPERSPHERICAL ENERGY TRANSFORMER FROM ITERATIVE ENERGY MINIMIZATION

In this section, we translate the proposed instantiation into two modified Hopfield energy functions defined on hyperspheres (see Appendix B for preliminaries and definition of Hopfield energy $E_{\mathrm{MCH}}$ in Eq. 16). Through iterative energy minimization, the architectural components of Transformer layers naturally arise under this framework. The overview is presented in Figure 2.

### 4.1    HYPERSPHERICAL ENERGY

Let $\boldsymbol{X} = [\boldsymbol{x}_1, \ldots, \boldsymbol{x}_N]$ denote a set of $N$ contextual token vectors, each $\boldsymbol{x}_i \in \mathbb{R}^d$. These tokens are projected into $H$ distinct subspaces via basis matrices $\boldsymbol{W} = [\boldsymbol{W}_1, \ldots, \boldsymbol{W}_H] \in \mathbb{R}^{d \times Hp}$, where each $\boldsymbol{W}_h \in \mathbb{R}^{d \times p}$ spans a $p$-dimensional subspace. Additionally, we define a second set of bases $\boldsymbol{D} = [\boldsymbol{d}_1, \ldots, \boldsymbol{d}_M] \in \mathbb{R}^{d \times M}$ to encode semantic directions in the original space. Unless otherwise specified, we assume these basis vectors are incoherent and span the full space, i.e., $Hp = M = d$.

#### 4.1.1    OVERCOMING TOKEN SYNCHRONIZATION VIA REPULSIVE DYNAMICS

Motivated by a recent argument that the contextual tokens lie on a low-dimensional manifold of their high-dimensional ambient space (Yu et al., 2023), we study the projection of tokens with bases $\boldsymbol{W}$; for a subspace spanned by $\boldsymbol{W}_h$, the latent representation of a token $\boldsymbol{x}_i$ can be written as

$$\boldsymbol{z}_i^h = \boldsymbol{W}_h^\top \boldsymbol{x}_i. \tag{2}$$

Canonical Hopfield energy $E_{\text{MCH}}$ tends to align vectors with stored patterns. This interaction often occurs between dynamic tokens and static patterns. However, in Transformers' self-attention, this interplay happens among all dynamic tokens simultaneously. Enforcing strict alignment among them risks collapsing representations into degenerate clusters, reducing expressiveness. This phenomenon has been observed empirically as oversmoothing (Chen et al., 2022; Wu et al., 2024b) or rank collapse (Dong et al., 2021), and theoretically characterized in (Geshkovski et al., 2023). It also relates to the synchronization effect in coupled systems (Acebrón et al., 2005; Miyato et al., 2025).

Therefore, to overcome this issue, we extend the Hopfield energy $E_{\text{MCH}}$ to model the repulsive force among tokens and quantify their distributional uniformity in each subspace, which serves as a surrogate of the uniformity measure in Eq. 1. For subspace $h$, this energy is given by

$$E_{\text{ATTN}}^h = \beta^{-1} \sum_{i=1}^{N} \log(\sum_{j=1}^{N} \exp\left(\beta(\boldsymbol{z}_i^h)^\top(\boldsymbol{z}_j^h)\right), \tag{3}$$

where $\beta$ is usually the inverse of temperature. Here we use the subscript $_{\text{ATTN}}$ as this energy will be shown to be related to the design of the attention layer, resembling that in Yu et al. (2023). Aggregating over all subspaces, the total energy that models interacting tokens would be

$$E_{\text{ATTN}} = \sum_{h=1}^{H} E_{\text{ATTN}}^h, \quad \text{subject to } \|\boldsymbol{W}_h^\top \boldsymbol{x}_i\|_2 = \sqrt{p}. \tag{4}$$

The constraint ensures that the dynamics take place on a low-dimensional hypersphere of radius $\sqrt{p}$. Minimizing $E_{\text{ATTN}}$ thus encourages token spread evenly on multiple hyperspheres, mitigating collapse and promoting distributional uniformity.[1]

### 4.1.2 SEMANTIC ALIGNMENT VIA ATTRACTION TO HIGH-DIMENSIONAL BASES

While the subspace projections separate to occupy more volume thus regularizing distribution, we seek to enrich the high-dimensional representations per se. From an information-theoretic perspective (Tishby et al., 2000; Tishby & Zaslavsky, 2015), effective representations require compressing uninformative redundancy while preserving salient information. Hence, in the original high-dimensional space, we encourage token alignment with a set of directions that contain knowledge from data to reduce entropy for minimal coding bits.

Motivated by empirical findings that feedforward layers in Transformers store much of their knowledge (Geva et al., 2021; Dar et al., 2023), we interpret the basis vectors $\boldsymbol{D}$ as the semantic directions. One surrogate function to implement this attractive energy for alignment in Eq. 1 is defined as

$$E_{\text{FF}} = -\frac{1}{2} \sum_{i=1}^{N} \sum_{m=1}^{M} \left(\text{ReLU}\left(\boldsymbol{d}_m^\top \boldsymbol{x}_i\right)\right)^2, \quad \text{subject to } \|\boldsymbol{D}^\top \boldsymbol{x}_i\|_2 = \sqrt{M}. \tag{5}$$

Here we use the subscript $_{\text{FF}}$ as this energy relates to the design of the feedforward layer. This energy favors alignment between tokens and those basis directions forming acute angles (as filtered by ReLU), while maintaining the hyperspherical constraint in the original space. Geometrically, each token is drawn toward a union of attractive half-spaces defined by $\boldsymbol{D}$. This could imply that each token may bind patterns combinatorially beyond the number of basis vectors defined by $\boldsymbol{D}$.

### 4.1.3 DUAL ENERGY ON THE HYPERSPHERE

By combining these two hyperspherical energy functions, we introduce a unified objective function that characterizes the functionality the Transformer layer represents:

$$\min_{\boldsymbol{x}_1,\ldots,\boldsymbol{x}_N \in \boldsymbol{X}} E(\boldsymbol{X}; \boldsymbol{W}, \boldsymbol{D}) = E_{\text{ATTN}} + E_{\text{FF}}, \tag{6}$$

$$\text{subject to} \quad \|\boldsymbol{W}_h^\top \boldsymbol{x}_i\|_2 = \sqrt{p}, \quad \|\boldsymbol{D}^\top \boldsymbol{x}_i\|_2 = \sqrt{M}, \quad i = 1,\ldots,N.$$

Iteratively minimizing this energy under spherical constraints induces the core architecture of Transformer layers: self-attention module arises from repulsive energy over subspaces and feedforward module arises from attractive energy in the ambient space. To solve optimization Eq. 6, we adopt an alternating minimization method by splitting it into sub-problems, following Yu et al. (2023).

---

[1]Its asymptotic convergence to uniformity on the sphere has been proven in Liu et al. (2018).

### 4.2 SYMMETRIC STRUCTURE INDUCED FROM ENERGY MINIMIZATION

#### 4.2.1 ATTENTION MODULE FROM UNIFORM ENERGY

To show how we have an attention module derived from minimizing hyperspherical energy $E_{\text{ATTN}}$ in Eq. 4, we first establish the differential equation that models the evolution of tokens' interactions:

$$
\begin{aligned}
\dot{\boldsymbol{X}} &= -\nabla_{\boldsymbol{X}} E_{\text{ATTN}} \\
&= -\sum_{h=1}^{H} \boldsymbol{W}_h \boldsymbol{W}_h^\top \boldsymbol{X} \left( \underbrace{\text{softmax}}_{\text{column-wise}} \left( \beta (\boldsymbol{W}_h^\top \boldsymbol{X})^\top (\boldsymbol{W}_h^\top \boldsymbol{X}) \right) + \underbrace{\text{softmax}}_{\text{row-wise}} \left( \beta (\boldsymbol{W}_h^\top \boldsymbol{X})^\top (\boldsymbol{W}_h^\top \boldsymbol{X}) \right) \right),
\end{aligned}
\tag{7}
$$

where $\beta = 1/\sqrt{p}$ as in vanilla Transformers. Derivations could be found in Appendix C.1.

The constraint on the low-dimensional hypersphere of radius $\sqrt{p}$ corresponds to $\text{RMSNorm}(\cdot)$, which bears resemblance to Query-Key Normalization (Henry et al., 2020), but here the normalization is applied after projection by the same query-key-value matrix. The projections in subspace $h$ onto the hypersphere thus read as

$$
\boldsymbol{Z}_{\text{RMS}}^h = \text{RMSNorm}(\boldsymbol{Z}^h) = \text{RMSNorm}(\boldsymbol{W}_h^\top \boldsymbol{X}).
\tag{8}
$$

By discretizing the differential equation Eq. 7 with step size $\alpha_t$ and maintaining the constraint Eq. 8, we obtain an self-attention module; let $[\boldsymbol{QK}]_{\text{RMS,t}} = \beta (\boldsymbol{Z}_{\text{RMS},t}^h)^\top (\boldsymbol{Z}_{\text{RMS},t}^h)$, then the update will be:

$$
\boldsymbol{X}_{t+1} = \boldsymbol{X}_t - \alpha_t \sum_{h=1}^{H} \left( \boldsymbol{W}_h \boldsymbol{Z}_{\text{RMS},t}^h \underbrace{\text{softmax}}_{\text{column-wise}} ([\boldsymbol{QK}]_{\text{RMS,t}}) + \boldsymbol{W}_h \boldsymbol{Z}_{\text{RMS},t}^h \underbrace{\text{softmax}}_{\text{row-wise}} ([\boldsymbol{QK}]_{\text{RMS,t}}) \right).
\tag{9}
$$

This update yields a doubly symmetric multi-head attention operator, where both the query-key dot product and attention weights are symmetric under row and column operations. This structure connects with formulations of Wasserstein gradient flows using doubly stochastic attention (Sander et al., 2022), grounding our energy-based interpretation.

#### 4.2.2 FEEDFORWARD MODULE FROM ALIGNMENT ENERGY

For the sub-problem of minimizing the alignment energy $E_{\text{FF}}$ in Eq. 5, we have a similar construction of the corresponding differential equation, with details deferred to Appendix C.2:

$$
\dot{\boldsymbol{X}} = -\nabla_{\boldsymbol{X}} E_{\text{FF}} = \boldsymbol{D} \, \text{ReLU} \left( \boldsymbol{D}^\top \boldsymbol{X} \right).
\tag{10}
$$

By further imposing the high-dimensional hyperspherical constraint via $\text{RMSNorm}$ with discretization step size $\gamma_t$, we can recover the feedforward layer that exhibits symmetry in the weight space:

$$
\boldsymbol{X}_{t+1} = \boldsymbol{X}_t + \gamma_t \boldsymbol{D} \, \text{ReLU} \left( \text{RMSNorm} \left( \boldsymbol{D}^\top \boldsymbol{X}_t \right) \right).
\tag{11}
$$

### 4.3 LEARNING ADAPTIVE STEP SIZE

To make the step sizes more flexible, we choose to learn their embedding with a neural network conditioned on the current iteration $t$ and the initial token $\boldsymbol{x}(0)$ (usually the output of the tokenizer):

$$
\alpha_t = \boldsymbol{\alpha}_\eta(t, \boldsymbol{x}(0)), \quad \gamma_t = \boldsymbol{\gamma}_\psi(t, \boldsymbol{x}(0)).
\tag{12}
$$

For each iteration, step size embeddings in Eq. 12 are applied channel-wise to each token, similar to techniques in Touvron et al. (2021); Peebles & Xie (2023) and detailed in Appendix D.1. We also adopt the zero-initialization of network parameters $\eta$ and $\psi$ from Bachlechner et al. (2021) to facilitate convergence when using larger iterations.

In summary, by combining all the components and techniques, we present the *Hyper-Spherical Energy Transformer* with only one layer of learnable parameters. This one-layer model is amenable to rigorous analysis and, as demonstrated later, has competitive performance with vanilla Transformer.

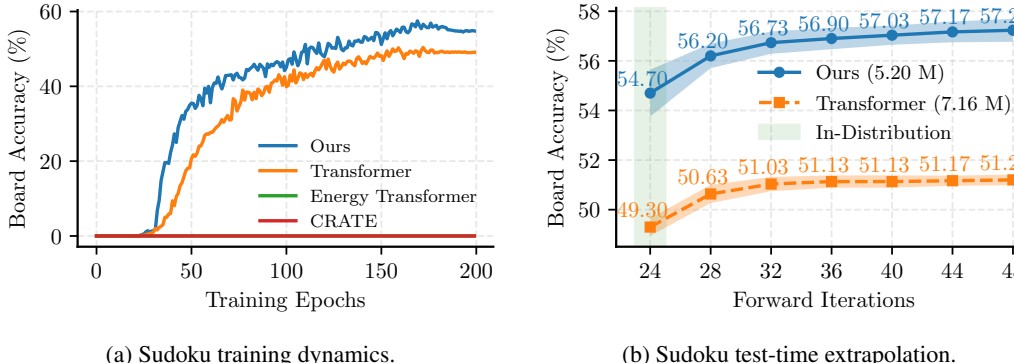

(a) Sudoku training dynamics.     (b) Sudoku test-time extrapolation.

Figure 3: *Left*: Training dynamics of different approaches. Our model achieves a superior training curve, while Energy Transformer and CRATE both fail to make non-trivial predictions (i.e., flat curves). *Right*: Test-time extrapolation w.r.t. forward iterations. Our model achieves better performance over five runs with fewer parameters, even when the iterations are beyond the training regime.

## 5    EXPERIMENT

In this section, we evaluate HYPER-SET against vanilla Transformer and other baselines on discriminative and generative tasks. For fairness, we remove biases and dropout, use the Pre-Norm style with $\mathrm{RMSNorm}$, and set the MLP ratio to 4 in Transformer. We use one-layer trainable parameters but vary the forward iterations for all models, including Transformers, unless otherwise specified.[2]

### 5.1    SOLVING SUDOKU

**Setups.**    We use the challenging dataset from Palm et al. (2018), featuring boards with only 17 to 34 known digits. We build on the code[3] from Yang et al. (2023) and follow the setting of training on 9k samples and evaluating on 1k. See Appendix D.2 for details.

**Extrapolation to Out-of-Distribution Iterations.**    Under identical experimental conditions, our model exhibits faster and superior training dynamics over Transformer, while Energy Transformer (Hoover et al., 2024) and white-box Transformer CRATE (Yu et al., 2023) both fail on this task, as shown in Figure 3a. It also outperforms Transformer for in-distribution evaluation (54.70% vs. 49.30%), i.e., using the same forward iterations for training and inference.

Recent efforts also explore test-time compute scaling to enhance reasoning (Schwarzschild et al., 2021; Bansal et al., 2022; Du et al., 2022; Banino et al., 2021), aiming to extrapolate the algorithms during training. Building on this idea, we increase test-time iterations up to $2\times$ of training ones. As shown in Figure 3b, our model scales more effectively than Transformer, with larger accuracy gains. We attribute this extrapolation to learned adaptive step sizes that preserve energy minimization. In practice, we also find that trainable positional encoding is vital for the extrapolation.

Sudoku could be challenging for some architectures to achieve non-zero results, as the $9\times9$ grid requires the exact digits to meet the constraints, which may be hard to learn without proper inductive bias. Even some foundation models have zero accuracy (Wang et al., 2025). A reasonable explanation for HYPER-SET's success, where CRATE and Energy Transformer fail, is that we have better modeling and more realistic assumptions about the design objective, which is based on as fundamental as maximum likelihood, making the architecture better aligned with the optimization procedure. This allows HYPER-SET to enjoy both the principled design and the expressivity of Transformer.

### 5.2    IMAGE CLASSIFICATION

**Setups & Results.**    We also compare HYPER-SET on CIFAR-10/100, ImageNet-100,[4] and the full ImageNet-1K against ViTs, CRATE (Yu et al., 2023) its variant CRATE-T that aims for more

---

[2]For instance, 12 iterations mean applying the layer repeatedly 12 times.

[3]https://github.com/azreasoners/recurrent_transformer

[4]We use a subset of ImageNet-1K from https://github.com/HobbitLong/CMC/blob/master/imagenet100.txt

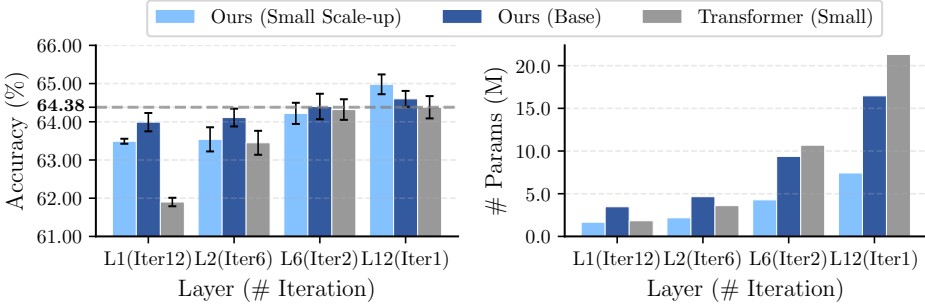

Figure 4: Top-1 accuracy (*Left*) and model parameters (*Right*) on CIFAR-100 with different layer-iteration trade-offs. Our model (Base) consistently surpasses Transformer, even its upper bound performance, with parameter efficiency. Error bars represent standard deviation over five runs.

faithful implementations (Hu et al., 2024c), and Energy Transformer (Hoover et al., 2024). Detailed setups are in Appendix D.3.

Table 1: Top-1 accuracy (%) for image classification with single-layer recurrent-depth models. Parameters are measured on ImageNet-1K. All models are trained from scratch on the listed datasets.

| Model | Width $d$ | # Params (M) | Dataset | | | |
|---|---|---|---|---|---|---|
| | | | CIFAR-10 | CIFAR-100 | IN-100 | IN-1K |
| Transformer | 384 | 2.38 | 89.90 | 61.89 | 69.44 | **66.94** |
| CRATE-T (Hu et al., 2024c) | 896 | 3.04 | 87.54 | 60.23 | 68.16 | 57.89 |
| CRATE (Yu et al., 2023) | 768 | 3.00 | 84.81 | 58.22 | 68.52 | 57.00 |
| Energy Transformer (Hoover et al., 2024) | 512 | 2.39 | 76.39 | 50.60 | 36.68 | 34.24 |
| HYPER-SET (Ours) | 512 | 2.39 | **90.11** | 63.41 | **70.16** | 62.76 |
| HYPER-SET (Ours) | 640 | 3.40 | 89.96 | **64.60** | 69.31 | 66.21 |

Table 1 shows that, under properly parameter-aligned settings, our model surpasses others on CIFAR-10/100 and ImageNet-100 but lags behind Transformer on large-scale ImageNet-1K. Notably, our architecture achieves a higher width-parameter ratio compared to Transformer, meaning more parameter-efficient under the same width.[5] When scaling up the width $d$, our model can narrow the performance gap to Transformer. This implies that our principled model suits better under a resource-constrained setting as its inherent structural biases could limit its scaling on large datasets.

**Layer-Iteration Trade-off.** So far, the classification is conducted using a one-layer model. A natural question is how well the model performs when stacking multiple layers with different parameters. To see this, we first train a Transformer with 12 layers—equivalent in effective depth to one layer with 12 iterations—as an upper bound. We then vary the number of distinct layers and their iterations while keeping total depth constant, effectively introducing flexibility to the basis vectors.

In Figure 4, our scaled-up `small` model has parameter efficiency across varied layer-iteration ratios, with this strength intensifying as more independent layers are trained. However, its architectural efficiency limits scalability beyond two layers. Scaling to the `Base` configurations enables our model to consistently outperform Transformer, exceeding the upper bound while retaining less parameters.

**Non-Recurrent Scenarios.** To demonstrate the practicality in the non-recurrent settings, which are more common in large-scale applications, we further provide comparisons in a 12-layer non-recurrent setting in Table 2, where we report results on CIFAR-10/100 with fine-tuning or training from scratch and on large-scale ImageNet-1K. The ImageNet-1K pretraining setup follows Table 1, with fine-tuning batch size 256, epochs 50, learning rate 1e-4, and weight decay 1e-5. Our model performs comparably as standard Transformers while being more parameter-efficient, and this parameter reduction constantly grows when stacking more distinct layers, as shown in Figure 4, thanks to our parsimonious design. This suggests that HYPER-SET potentially preserves the benefit of

---

[5]We compare the parameter efficiency and computational cost in Appendix D.6.

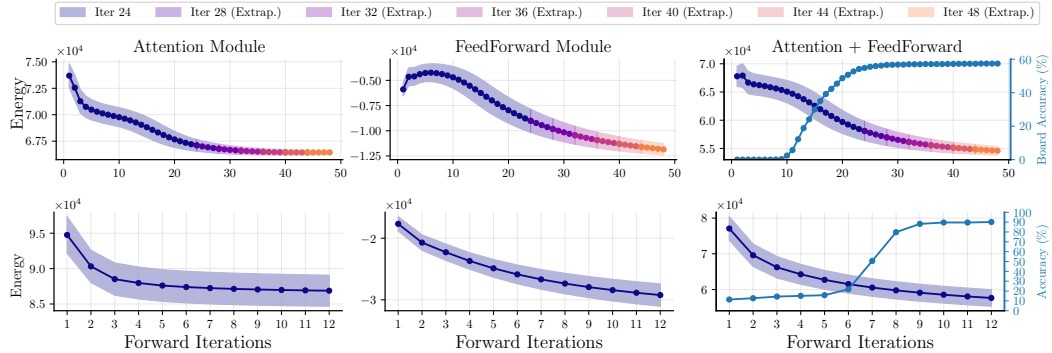

Figure 5: The attention and feedforward energy decrease on Sudoku (*Top*) and CIFAR-10 (*Down*) even without sign constraints on step sizes. This suggests the layer aligns well with the optimization objective. Normalization is first applied to meet constraints in Eq. 6 before computing the energy. In the right subfigure, the decrease in the overall energy corresponds to the increase in performance.

scaling in Transformers when extended practically, while being more useful in resource-constrained settings where model size matters, further underscoring its promise in practical scenarios.

Table 2: Top-1 accuracy (%) for image classification with 12-layer non-recurrent-depth models. Parameters are measured on ImageNet-1K. [†] means first pretraining on ImageNet-1K and then fine-tuning on CIFAR-10/100.

| Model | Width $d$ | # Params (M) | Dataset | | | | |
|---|---|---|---|---|---|---|---|
| | | | CIFAR-10 | CIFAR-100 | CIFAR-10[†] | CIFAR-100[†] | IN-1K |
| Transformer | 384 | 21.86 | 87.44 | 62.84 | **96.95** | **83.10** | **67.90** |
| CRATE (Yu et al., 2023) | 512 | 10.28 | 88.76 | 63.94 | 94.18 | 77.39 | 60.69 |
| HYPER-SET (Ours) | 512 | 8.17 | **88.82** | **64.98** | 95.76 | 80.89 | 66.26 |
| HYPER-SET (Ours) | 768 | 17.56 | 88.53 | 64.16 | 96.47 | 82.60 | 67.20 |

## 5.3 MASKED IMAGE MODELING

**Setups & Results.** Masked image modeling has recently regained its traction for autoregressive generation (Li et al., 2023; 2024), framed as recovering images from 100% masking. Due to its high computational demand, we attempt to demonstrate the power of our one-layer model specifically for image reconstruction on ImageNet-100. We build on prior work (Chang et al., 2022) and use the open-source repository.[6] Concrete settings are in Appendix D.4, with additional results and visualization in Appendix E.

Table 3 unveils that, under the same number of iterations, our model significantly reduces parameters but lags behind Transformer on all metrics. If we further increase its iterations and the width of feedforward module $M$ to $8d$, it can fill in the performance gap but at the cost of more computation.

Table 3: Comparisons of masked image modeling performance on ImageNet-100 (5k). While our model is more compact, scaling the feedforward module width ($M$) allows it to match Transformer performance. All models use a single layer.

| Model | Iters | FF Ratio $M$ | # Params (M) | PSNR (↑) | SSIM (↑) | MS-SSIM (↑) | LPIPS (↓) | FID (↓) |
|---|---|---|---|---|---|---|---|---|
| Transformer | 12 | $4d$ | 8.85 | 15.953 | **0.417** | **0.599** | **0.327** | **43.428** |
| HYPER-SET (Ours) | 12 | $1d$ | 3.94 | 15.713 | 0.411 | 0.576 | 0.358 | 59.841 |
| HYPER-SET (Ours) | 24 | $8d$ | 8.07 | **15.955** | 0.417 | 0.596 | 0.332 | 45.174 |

## 5.4 ENERGY EVOLUTION, EFFECTIVE RANK AND AVERAGE ANGLE

Figure 5 shows energy trajectories of the attention ($E_{\text{ATTN}}$) and feedforward module ($E_{\text{FF}}$). Even without a positive threshold for step sizes $\alpha_t$ and $\gamma_t$, the energy on Sudoku still decreases within

---

[6]https://github.com/valeoai/Maskgit-pytorch

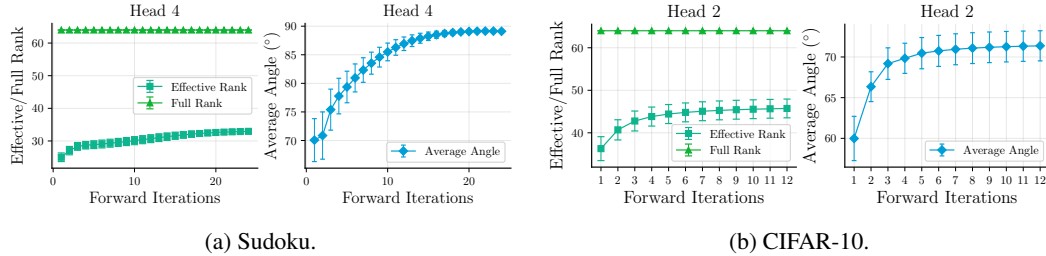

(a) Sudoku.  (b) CIFAR-10.

Figure 6: The effective rank and average angle of tokens projected to one subspace gradually increase, suggesting a larger volume spanned by these tokens. Results are from Sudoku test dataset (Palm et al., 2018) (*Left*) and CIFAR-10 validation set (*Right*).

training iterations and extrapolates smoothly beyond them, indicating strong generalization of learned step sizes. On CIFAR-10, our designed energy exhibits a monotonic decline as well.

To verify our subspace uniformity objective, we track two metrics—*effective rank* and *average angle*—defined in Appendix D.5. As shown in Figure 6, the subspace effective rank steadily increases while the full rank remains unchanged. Meanwhile, average angles between tokens approach orthogonality, aligning with our goal to prevent entropy collapse. Full results and comparisons with parameter-sharing Transformer are in Appendix F.

Table 4: Alternative design instantiations on key components measured by top-1 accuracy (%).

| Components | Alternative Designs | CIFAR-10 | CIFAR-100 |
|---|---|---|---|
| $E_{\text{ATTN}}$ | Bi-Softmax Attention (Default) | **90.11** | **63.41** |
| | Sigmoid Attention | 85.93 | 59.72 |
| | Linear Attention | 84.88 | 56.97 |
| $E_{\text{FF}}$ | ReLU FF (Default) | **90.11** | **63.41** |
| | Softmax FF | 88.20 | 62.44 |
| | Gated FF | 84.99 | 59.29 |
| Step Size | Learned Step Size (Default) | **90.11** | **63.41** |
| | $\alpha_t = \gamma_t = 0.5$ | 25.81 | 57.92 |
| | $\alpha_t = \gamma_t = 0.1$ | 81.45 | 58.29 |

Table 5: ImageNet-100 accuracy (%) of HYPER-SET under different matrix rank ($r$) in LoRA. Depth-wise LoRA introduces representation flexibility at each iteration.

| Model | # Params (M) | Accuracy |
|---|---|---|
| HYPER-SET (Ours) | 1.93 | 70.16 |
| + depth-wise LoRA ($r = 4$) | 2.03 | 70.36 |
| + depth-wise LoRA ($r = 8$) | 2.13 | 70.40 |
| + depth-wise LoRA ($r = 16$) | 2.33 | 70.56 |
| + depth-wise LoRA ($r = 32$) | 2.72 | **72.20** |

### 5.5 ALTERNATIVE DESIGNS AND SCALABILITY

A key strength of our formulation Eq. 1 lies in its generality—it supports a broad spectrum of model variants through alternative energy functions. For example, replacing the attention energy with a kernel-based function yields novel attention mechanisms, including linear attention. Similarly, gating in feedforward layers naturally arises by generalizing the feedforward energy. Table 4 shows the performance of these variants and results with different fixed step sizes, with details in Appendix G.1

To improve scalability with compact parameterizations, we introduce a lightweight extension inspired by Bae et al. (2025), where learnable low-rank adapters are added at each forward iteration to modulate shared weights. This depth-wise adaptation, shown in Table 5, enhances performance without significantly increasing parameter count. Setups and additional scaling results on image and text modalities are included in Appendix G.2.

## 6 CONCLUSION

We present HYPER-SET, a Transformer architecture designed via iterative optimization of hyperspherical energy functions, bridging energy-based learning and practical model design. By formulating dual energy on the hypersphere under a general principle derived from maximum likelihood, HYPER-SET pursues distributional uniformity in the low-dimensional subspaces while promoting directional alignment with bases in the original high-dimensional space, constructing core Transformer components with intrinsic interpretability. Empirically, HYPER-SET matches or surpasses vanilla Transformers across diverse tasks with fewer parameters. Beyond a single architecture, our framework enables flexible design choices and scalable variants. This work contributes towards principled, more describable, and economical Transformer designs that are both theoretically motivated and practically effective. We discuss limitations and future directions in Appendix H.

## ACKNOWLEDGMENTS

This work is supported by the Research Grants Council (RGC) of the Hong Kong SAR under the General Research Fund (17203023), the Collaborative Research Fund (C5052-23G), and the NSFC/RGC Collaborative Research Scheme (CRS_HKU703/24). The research work described in this paper was conducted in the JC STEM Lab of Multimedia and Machine Learning funded by The Hong Kong Jockey Club Charities Trust. Difan Zou acknowledges the support from NSFC 62306252, Hong Kong ECS award 27309624, Guangdong NSF 2024A1515012444, and the fund from SLAI.

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

# A   THEORETICAL JUSTIFICATION OF MOTIVATION

We provide here a theoretical foundation for the objective in Eq. 1, showing how it arises naturally from a maximum likelihood estimation (MLE) framework under mild assumptions.

Let $\boldsymbol{x} \in \mathbb{R}^d$ be a random vector (considered as a token in the context of Transformer) in a high-dimensional representation space with a probability distribution $p(\boldsymbol{x})$. Let $\{\boldsymbol{z}^h\}_{h=1}^H$ be a set of random vectors in low-dimensional latent spaces $\mathbb{R}^p$ $(p < d)$ with distinct support, following a prior joint probability distribution $p(z^1, \ldots, z^H)$. We formulate information processing in the forward pass of neural networks as maximum likelihood estimation:

$$\max_{\boldsymbol{x}} \quad \mathbb{E}_{(\boldsymbol{z}^1, \ldots, \boldsymbol{z}^H) \sim p(\boldsymbol{z}^1, \ldots, \boldsymbol{z}^H)} \left[ \log p(\boldsymbol{x}, \boldsymbol{z}^1, \ldots, \boldsymbol{z}^H; \theta, \phi) \right], \tag{13}$$

where $\theta$ and $\phi$ are parameters of the high- and low-dimensional encodings, respectively.

To make this optimization more tractable, we make the following basic and practical assumptions:

**Assumption 1.** *The random vectors $\boldsymbol{z}^1, \ldots, \boldsymbol{z}^H$ are independent and follow the identical distribution $p(\boldsymbol{z})$ in distinct latent spaces, i.e., $p(\boldsymbol{z}^1, \ldots, \boldsymbol{z}^H) = \Pi_{h=1}^H p(\boldsymbol{z}^h)$ and $p(\boldsymbol{z}^1) = \cdots = p(\boldsymbol{z}^H) = p(\boldsymbol{z})$.*

**Assumption 2.** *The prior distribution $p(\boldsymbol{z})$ is a uniform distribution with support on a hypersphere $\mathbb{S}^{p-1}$.*

**Assumption 3.** *The random vectors $(\boldsymbol{z}^1, \ldots, \boldsymbol{z}^H) \sim p_\phi(\boldsymbol{z}^1, \ldots, \boldsymbol{z}^H \mid \boldsymbol{x})$ from the posterior distribution are conditionally independent, i.e., $p_\phi(\boldsymbol{z}^1, \ldots, \boldsymbol{z}^H \mid \boldsymbol{x}) = \Pi_{h=1}^H p_\phi(\boldsymbol{z}^h \mid \boldsymbol{x})$.*

Assumption 2 of hyperspherical uniform distribution can be perceived to function as regularization on the latent representations to preserve maximum entropy and avoid representational collapse, which has been adopted to enhance auto-encoding (Xu & Durrett, 2018; Davidson et al., 2018). Under the above basic and practical assumptions, the MLE objective can be reformulated as:

$$\begin{aligned}
\max_{\boldsymbol{x}} &\ \mathbb{E}_{(\boldsymbol{z}^1, \ldots, \boldsymbol{z}^H) \sim p(\boldsymbol{z}^1, \ldots, \boldsymbol{z}^H)} \left[ \log p(\boldsymbol{x}, \boldsymbol{z}^1, \ldots, \boldsymbol{z}^H; \theta, \phi) \right] \\
&= \mathbb{E}_{(\boldsymbol{z}^1, \ldots, \boldsymbol{z}^H) \sim p(\boldsymbol{z})} \left[ \log p_\phi(\boldsymbol{z}^1, \ldots, \boldsymbol{z}^H \mid \boldsymbol{x}) \right] + \mathbb{E}_{(\boldsymbol{z}^1, \ldots, \boldsymbol{z}^H) \sim p(\boldsymbol{z})} \left[ \log p_\theta(\boldsymbol{x}) \right] \\
&= \sum_{h=1}^H \mathbb{E}_{\boldsymbol{z}^h \sim p(\boldsymbol{z})} \left[ \log p_\phi(\boldsymbol{z}^h \mid \boldsymbol{x}) \right] + \log p_\theta(\boldsymbol{x}) \\
&= \sum_{h=1}^H \mathbb{E}_{\boldsymbol{z}^h \sim p(\boldsymbol{z})} \left[ \log \frac{p_\phi(\boldsymbol{z}^h \mid \boldsymbol{x})}{p(\boldsymbol{z}^h)} \right] + \sum_{h=1}^H \mathbb{E}_{\boldsymbol{z}^h \sim p(\boldsymbol{z})} \left[ \log p(\boldsymbol{z}^h) \right] + \log p_\theta(\boldsymbol{x}) \\
&= \sum_{h=1}^H \mathbb{E}_{\boldsymbol{z}^h \sim p(\boldsymbol{z})} \left[ \log \frac{p_\phi(\boldsymbol{z}^h \mid \boldsymbol{x})}{p(\boldsymbol{z})} \right] + \sum_{h=1}^H \mathbb{E}_{\boldsymbol{z} \sim p(\boldsymbol{z})} \left[ \log p(\boldsymbol{z}) \right] + \log p_\theta(\boldsymbol{x}) \\
&= -\sum_{h=1}^H D_{\mathrm{KL}}(p(\boldsymbol{z}) \| p_\phi(\boldsymbol{z}^h \mid \boldsymbol{x})) - H \times \mathcal{H}(p(\boldsymbol{z})) + \log p_\theta(\boldsymbol{x}), \tag{14}
\end{aligned}$$

where $D_{\mathrm{KL}}(\cdot \| \cdot)$ denotes Kullback-Leibler (KL) divergence and $\mathcal{H}(\cdot)$ means differential entropy. As the second term on entropy in Eq. 14 does not depend on variable $\boldsymbol{x}$, this objective ultimately reduces to Eq. 1 which we restate here for completeness:

$$\min_{\boldsymbol{x}} \quad \sum_{h=1}^H \underbrace{D_{\mathrm{KL}}\left( p(\boldsymbol{z}) \| p_\phi(\boldsymbol{z}^h \mid \boldsymbol{x}) \right)}_{\text{uniformity}} \underbrace{- \log(p_\theta(\boldsymbol{x}))}_{\text{alignment}}.$$

The first term encourages the posterior $p_\phi(\boldsymbol{z}^h \mid \boldsymbol{x})$ defined on the vector $\boldsymbol{z}^h \in \mathbb{R}^p$ in latent space, which can be implemented by a transformation parameterized by $\phi$, to approximate a uniform distribution on a hypersphere. To see why the second term implies alignment, suppose the distribution $p_\theta(\boldsymbol{x})$ is parameterized as a mixture of $M$ von Mises–Fisher (vMF) distributions[7] with equal mixing

---

[7]https://en.wikipedia.org/wiki/Von_Mises–Fisher_distribution

coefficients:

$$-\log(p_\theta(\boldsymbol{x})) = -\log\left(\frac{1}{M}\sum_{m=1}^{M} f(\boldsymbol{x};\boldsymbol{\mu}_m,\kappa_m)\right) = -\log\left(\frac{1}{M}\sum_{m=1}^{M} C_d(\kappa_m)\exp(\kappa_m\boldsymbol{\mu}_m^\top\boldsymbol{x})\right),$$
(15)

where $\boldsymbol{\mu}_m \in \mathbb{S}^{d-1}$ denotes mean direction on $(d-1)$-dimensional unit sphere and $\kappa_m \geq 0$ is the concentration parameter while $C_d(\kappa_m)$ is the normalization constant; therefore, finding the vector $\boldsymbol{x}$ that minimizes this negative log-probability Eq. 15 equals finding maximal inner product $\boldsymbol{\mu}_m^\top\boldsymbol{x}$, thus aiming for directional alignment. In practice, the mean direction $\boldsymbol{\mu}_m$ is learned by backpropagation and consequently contains certain statistical properties from the data.

In summary, the objective Eq. 1 suggests that for a representation vector $\boldsymbol{x} \in \mathbb{R}^d$, the forward dynamics can be characterized by two complementary properties:

- **Mode Seeking**: Achieving semantic alignment with directional vectors encapsulating specific information derived from data in the **high-dimensional space**.

- **Mass Covering**: Maximally preserving the entropy embedded via regularizing distributional uniformity in the **low-dimensional space**.

These principles underpin our design of token dynamics, and we propose to use energy functions to quantify these two properties as instantiations that can induce various Transformer-based models.

## B PRELIMINARIES

### B.1 HOPFIELD NETWORKS

Given a network with $N$ neurons $\boldsymbol{x} = [x_1,\ldots,x_N]$ that take binary values, the temporal evolution dynamics of these neurons are determined by a scalar-value energy function:

$$E = -\frac{1}{2}\sum_{i,j}\omega_{ij}x_ix_j = -\frac{1}{2}\boldsymbol{x}^\top\boldsymbol{W}\boldsymbol{x}, \quad x_i, x_j \in \{+1, -1\}$$

where $\omega_{ij}$ represents the strength of connectivity between node $x_i$ and $x_j$, and the connectivity is assumed to be symmetric, i.e., $\omega_{ij} = \omega_{ji}$. We can further rewrite $\boldsymbol{W} = \sum_{i=1}^{P}\boldsymbol{\xi}_i\boldsymbol{\xi}_i^\top$ as a set of patterns to be stored. The update rule of each node to retrieve the most relevant pattern follows the Hebbian learning rule used in neuroscience:

$$\boldsymbol{x}_{t+1} = \text{sign}(\boldsymbol{W}\boldsymbol{x}_t) = \text{sign}\left(\sum_{i=1}^{P}\boldsymbol{\xi}_i\boldsymbol{\xi}_i^\top\boldsymbol{x}_t\right).$$

This update rule tends to minimize the energy function with retrieved patterns as its attractor. It is an embodiment of the idea of "Neurons that fire together wire together.": If two neurons connect ($\omega_{ij} > 0$), then they should have the same state ($+1$ for active and $-1$ for dead). The number of patterns the network can store and retrieve is $\mathcal{O}(N)$.

### B.2 MODERN CONTINUOUS HOPFIELD NETWORKS

To overcome the limitation of linear storage capacity, modern Hopfield networks, also known as Dense Associative Memory (Krotov & Hopfield, 2016), introduce nonlinearity in the energy and the update of neurons' states and make them suitable for continuous variables:

$$E = -\frac{1}{2}\sum_{i=1}^{P} f\left(\boldsymbol{\xi}_i^\top\boldsymbol{x}\right), \ \boldsymbol{x}_{t+1} = \tanh\left(\sum_{i=1}^{P}\boldsymbol{\xi}_i f'\left(\boldsymbol{\xi}_i^\top\boldsymbol{x}_t\right)\right),$$

where $\tanh(\cdot)$ is to ensure the neurons' states are constrained to the interval $[-1, 1]$ so that the energy is bounded from below. Depending on the form of $f$, the network could have power or exponential storage capacity. If we set $f(x) = x^2$, this reduces to the traditional networks with linear capacity.

If we further make modifications to the non-linearity in the energy function with $\mathrm{logsumexp}(\cdot)$, which is inspired by contrastive normalization, we can define the Modern Continuous Hopfield (MCH) energy function with a quadratic regularization term on $\boldsymbol{x}$:

$$E_{\mathrm{MCH}} = -\log\left(\sum_{i=1}^{P} \exp\left(\boldsymbol{\xi_i}^\top \boldsymbol{x}\right)\right) + \frac{1}{2}\boldsymbol{x}^\top \boldsymbol{x}. \tag{16}$$

By leveraging the concave-convex procedure (Yuille & Rangarajan, 2003), the update could be written as

$$\boldsymbol{x}_{t+1} = \Xi\,\mathrm{softmax}(\Xi^\top \boldsymbol{x}_t),$$

where $\Xi = [\xi_1, \ldots, \xi_P] \in \mathbb{R}^{N \times P}$. This formulation has proven to converge to stationary points of the energy function $E_{\mathrm{MCH}}$, and is linked to the key-value memory similar to the attention mechanism (Ramsauer et al., 2021). Notice that this update rule is essentially the cross-attention given a query vector $\boldsymbol{x}$ and can only describe the independent evolution of that vector. It fails to faithfully cover the parallel interactions between contextual tokens in the self-attention adopted in the GPT or BERT style Transformers.

The construction of the modern continuous Hopfield energy and update rule can also be carried out from a biologically plausible view by extending the network with hidden neurons and establishing a group of coupled differential equations. We refer the readers to Krotov & Hopfield (2021); Krotov (2023) for more details.

## C    DERIVATION

### C.1    DERIVATION OF THE GRADIENT OF $E_{\mathrm{ATTN}}$

$$
\begin{aligned}
\dot{\boldsymbol{x}}_k &= -\nabla_{\boldsymbol{x}_k} E_{\mathrm{ATTN}} \\
&= -\sum_{h=1}^{H}\left(\frac{\sum_{j=1}^{N}\boldsymbol{W}_h\boldsymbol{W}_h^\top\boldsymbol{x}_j\exp\left(\beta(\boldsymbol{W}_h^\top\boldsymbol{x}_k)^\top(\boldsymbol{W}_h^\top\boldsymbol{x}_j)\right)}{\sum_{j=1}^{N}\exp\left(\beta(\boldsymbol{W}_h^\top\boldsymbol{x}_k)^\top(\boldsymbol{W}_h^\top\boldsymbol{x}_j)\right)} + \sum_{i=1}^{N}\frac{\boldsymbol{W}_h\boldsymbol{W}_h^\top\boldsymbol{x}_i\exp\left(\beta(\boldsymbol{W}_h^\top\boldsymbol{x}_i)^\top(\boldsymbol{W}_h^\top\boldsymbol{x}_k)\right)}{\sum_{j=1}^{N}\exp\left(\beta(\boldsymbol{W}_h^\top\boldsymbol{x}_i)^\top(\boldsymbol{W}_h^\top\boldsymbol{x}_j)\right)}\right) \\
&= -\sum_{h=1}^{H}\left(\boldsymbol{W}_h\boldsymbol{W}_h^\top[\boldsymbol{x}_1,\ldots,\boldsymbol{x}_N]\begin{bmatrix}\exp\left(\beta(\boldsymbol{W}_h^\top\boldsymbol{x}_k)^\top(\boldsymbol{W}_h^\top\boldsymbol{x}_1)\right)\\ \vdots \\ \exp\left(\beta(\boldsymbol{W}_h^\top\boldsymbol{x}_k)^\top(\boldsymbol{W}_h^\top\boldsymbol{x}_N)\right)\end{bmatrix}\bigg/\sum_{j=1}^{N}\exp\left(\beta(\boldsymbol{W}_h^\top\boldsymbol{x}_k)^\top(\boldsymbol{W}_h^\top\boldsymbol{x}_j)\right) + \right. \\
&\qquad\left. \sum_{i=1}^{N}\boldsymbol{W}_h\boldsymbol{W}_h^\top\boldsymbol{x}_i\begin{bmatrix}\exp\left(\beta(\boldsymbol{W}_h^\top\boldsymbol{x}_1)^\top(\boldsymbol{W}_h^\top\boldsymbol{x}_i)\right)\big/\sum_{j=1}^{N}\exp\left(\beta(\boldsymbol{W}_h^\top\boldsymbol{x}_i)^\top(\boldsymbol{W}_h^\top\boldsymbol{x}_j)\right)\\ \vdots \\ \exp\left(\beta(\boldsymbol{W}_h^\top\boldsymbol{x}_N)^\top(\boldsymbol{W}_h^\top\boldsymbol{x}_i)\right)\big/\sum_{j=1}^{N}\exp\left(\beta(\boldsymbol{W}_h^\top\boldsymbol{x}_i)^\top(\boldsymbol{W}_h^\top\boldsymbol{x}_j)\right)\end{bmatrix}_k\right) \\
&= -\sum_{h=1}^{H}\left(\boldsymbol{W}_h\boldsymbol{W}_h^\top\boldsymbol{X}\underbrace{\mathrm{softmax}}_{\text{column}}\left(\beta(\boldsymbol{W}_h^\top\boldsymbol{X})^\top(\boldsymbol{W}_h^\top\boldsymbol{x}_k)\right) + \sum_{i=1}^{N}\boldsymbol{W}_h\boldsymbol{W}_h^\top\boldsymbol{x}_i\underbrace{\mathrm{softmax}}_{\text{column}}\left(\beta(\boldsymbol{W}_h^\top\boldsymbol{X})^\top(\boldsymbol{W}_h^\top\boldsymbol{x}_i)\right)_k\right) \\
&= -\sum_{h=1}^{H}\left(\boldsymbol{W}_h\boldsymbol{W}_h^\top\boldsymbol{X}\underbrace{\mathrm{softmax}}_{\text{column}}\left(\beta(\boldsymbol{W}_h^\top\boldsymbol{X})^\top(\boldsymbol{W}_h^\top\boldsymbol{x}_k)\right) + \boldsymbol{W}_h\boldsymbol{W}_h^\top\boldsymbol{X}\underbrace{\mathrm{softmax}}_{\text{column}}\left(\beta(\boldsymbol{W}_h^\top\boldsymbol{X})^\top(\boldsymbol{W}_h^\top\boldsymbol{X})\right)_{[k,:]}\right) \\
&= -\sum_{h=1}^{H}\left(\boldsymbol{W}_h\boldsymbol{W}_h^\top\boldsymbol{X}\underbrace{\mathrm{softmax}}_{\text{column}}\left(\beta(\boldsymbol{W}_h^\top\boldsymbol{X})^\top(\boldsymbol{W}_h^\top\boldsymbol{x}_k)\right) + \boldsymbol{W}_h\boldsymbol{W}_h^\top\boldsymbol{X}\underbrace{\mathrm{softmax}}_{\text{row}}\left(\beta(\boldsymbol{W}_h^\top\boldsymbol{X})^\top(\boldsymbol{W}_h^\top\boldsymbol{X})\right)_{[:,k]}\right) \\
&= -\sum_{h=1}^{H}\left(\boldsymbol{W}_h\boldsymbol{W}_h^\top\boldsymbol{X}\underbrace{\mathrm{softmax}}_{\text{column}}\left(\beta(\boldsymbol{W}_h^\top\boldsymbol{X})^\top(\boldsymbol{W}_h^\top\boldsymbol{x}_k)\right) + \boldsymbol{W}_h\boldsymbol{W}_h^\top\boldsymbol{X}\underbrace{\mathrm{softmax}}_{\text{row}}\left(\beta(\boldsymbol{W}_h^\top\boldsymbol{X})^\top(\boldsymbol{W}_h^\top\boldsymbol{X})\right)_{[:,k]}\right)
\end{aligned}
$$

$$\dot{\boldsymbol{X}} = [\dot{\boldsymbol{x}}_1, \ldots, \dot{\boldsymbol{x}}_N]$$
$$= -\nabla_{\boldsymbol{X}} E_{\text{ATTN}}$$
$$= -\left( (\boldsymbol{W}\boldsymbol{W}^\top \boldsymbol{X} \underbrace{\text{softmax}}_{\text{column-wise}} (\beta(\boldsymbol{W}^\top \boldsymbol{X})^\top (\boldsymbol{W}^\top \boldsymbol{X})) + \boldsymbol{W}\boldsymbol{W}^\top \boldsymbol{X} \underbrace{\text{softmax}}_{\text{row-wise}} (\beta(\boldsymbol{W}^\top \boldsymbol{X})^\top (\boldsymbol{W}^\top \boldsymbol{X})) \right)$$

## C.2 DERIVATION OF THE GRADIENT OF $E_{\text{FF}}$

$$\dot{\boldsymbol{x}}_k = -\nabla_{\boldsymbol{x}_k} E_{\text{FF}}$$
$$= \sum_{m=1}^{M} \text{ReLU}(\boldsymbol{d}_m^\top \boldsymbol{x}_k) \cdot \mathbb{I}(\boldsymbol{d}_m^\top \boldsymbol{x}_k > 0) \cdot \boldsymbol{d}_m$$
$$= \sum_{m=1}^{M} \text{ReLU}(\boldsymbol{d}_m^\top \boldsymbol{x}_k) \boldsymbol{d}_m$$
$$= [\boldsymbol{d}_1, \ldots, \boldsymbol{d}_M] \begin{bmatrix} \text{ReLU}(\boldsymbol{d}_1^\top \boldsymbol{x}_k) \\ \vdots \\ \text{ReLU}(\boldsymbol{d}_M^\top \boldsymbol{x}_k) \end{bmatrix}$$
$$= \boldsymbol{D} \, \text{ReLU}(\boldsymbol{D}^\top \boldsymbol{x}_k)$$

$$\dot{\boldsymbol{X}} = [\dot{\boldsymbol{x}}_1, \ldots, \dot{\boldsymbol{x}}_N] = -\nabla_{\boldsymbol{X}} E_{\text{FF}} = \boldsymbol{D} \, \text{ReLU}(\boldsymbol{D}^\top \boldsymbol{X})$$

## D DETAILED EXPERIMENTAL SETUPS AND MODEL CONFIGURATIONS

### D.1 NETWORK TO LEARN ADAPTIVE STEP SIZES

We propose to learn adaptive step sizes $\boldsymbol{\alpha}_t, \boldsymbol{\gamma}_t \in \mathbb{R}^d$ given the current iteration $t \in [1, L]$, where $L$ is the iteration number of the layer with unique parameters, and the initial token $\boldsymbol{x}(0) \in \mathbb{R}^d$, using the network shown in Figure 7 and configurations in Table 6.

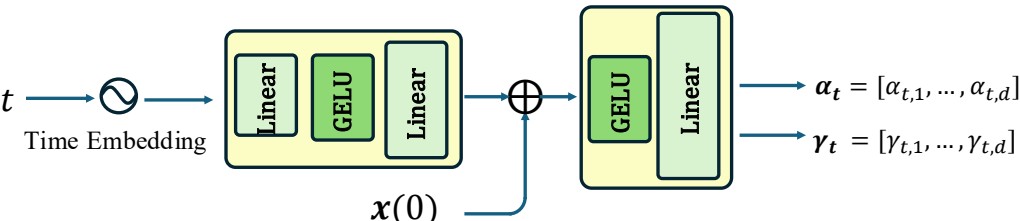

Figure 7: Illustration of time embedding conditioned on the input to learn adaptive step size.

Table 6: Model configurations of network to learn adaptive step sizes.

| Layer | Configuration |
|---|---|
| Time embedding | 512 |
| Linear | $512 \times d$ |
| GELU | – |
| Linear | $d \times d$ |
| GELU | – |
| Linear | $d \times 2d$ |

## D.2 Solving Sudoku

Solving a Sudoku puzzle requires filling a $9 \times 9$ board, with some digits (1-9) known and unknown entries marked as 0. The unknown entries must be filled with digits perfectly such that the board satisfies a certain rule, which can be seen as a logical reasoning task (Wang et al., 2019). We tackle this puzzle by predicting the digits to fill in, conditioned on the given digits. It can be viewed as a simplified masked modeling on synthetic data. Cross-entropy loss is computed exclusively on unknown entries. We train all models with 200 epochs, 16 batch size, AdamW (Loshchilov & Hutter, 2019) with 0.1 weight decay, and learning rate from $1 \times 10^{-4}$ with cosine decay. The hidden dimension is set to 768. Tables 7 and 8 show the training recipe and model configurations for solving Sudoku. We train the model with 24 iterations and can evaluate beyond these iterations.

Table 7: Training recipe for solving Sudoku.

| Hyperparameter | Value |
|---|---|
| Epochs | 200 |
| Batch size | 16 |
| # GPU | $1 \times$ NVIDIA 3090 |
| # Training samples | 9,000 |
| # Evaluating samples | 1,000 |
| Optimizer | AdamW |
| $\beta_1, \beta_2$ | 0.9, 0.95 |
| Weight decay | 0.1 |
| Learning rate (lr) | $1 \times 10^{-4}$ |
| LR decay | Cosine |
| Gradient clipping | 1.0 |

Table 8: Model configurations for solving Sudoku.

| Configuration | Value |
|---|---|
| Vocabulary size | 10 |
| Layer | 1 |
| Iterations $L$ | 24 |
| Hidden dimension $d$ | 768 |
| Feedforward ratio $M$ | $4d$ |
| Number of heads $H$ | 12 |
| Positional encoding | Learnable |
| Time embedding condition | $X_0$ |
| Time embedding frequency | 512 |
| Total parameters | 5.20 M |

## D.3 Image Classification

Tables 9 and 10 present the training recipe and model configurations for image classification on CIFAR-10/100, while Tables 11 and 12 show those on ImageNet-100/1K. All models are trained with a learnable class token [CLS]. In practice, we use absolute sinusoidal positional encoding and adopt conditioning on $X_t$ for performance reasons. Table 13 lists the configurations of different sizes, and it applies to other tasks as well.

Table 9: Training recipe for image classification on CIFAR-10/100.

| Hypersphere | Value |
|---|---|
| Epochs | 200 |
| Batch size | 128 |
| # GPU | $1 \times$ NVIDIA 3090 |
| # Training samples | 50,000 |
| # Evaluating samples | 10,000 |
| Optimizer | Adam |
| $\beta_1, \beta_2$ | 0.9, 0.999 |
| Weight decay | $5 \times 10^{-5}$ |
| Max learning rate (lr) | $1 \times 10^{-3}$ |
| Min learning rate (lr) | $1 \times 10^{-5}$ |
| LR decay | Cosine |
| Warmup epochs | 5 |
| Input size | 32 |

Table 10: Model configurations (Small Scaled-up) for image classification on CIFAR-10/100.

| Configuration | Value |
|---|---|
| Patch size | 8 |
| Layer | 1 |
| Iterations $L$ | 12 |
| Hidden dimension $d$ | 512 |
| Feedforward ratio $M$ | $d$ |
| Number of heads $H$ | 8 |
| Positional encoding | Sinusoidal |
| Time embedding condition | $X_t$ |
| Time embedding frequency | 512 |

## D.4 Masked Image Modeling

We follow Chang et al. (2022) using VQ-VAE to tokenize the images to $16 \times 16$ latent code with the codebook size of 1024 after resizing the input to $256 \times 256$. The masking ratio is randomly

Table 11: Training recipe for image classification on ImageNet-100/1K.

| Hyperparameter | Value |
|---|---|
| Epochs | 200 |
| Batch size | 256 |
| # GPU | $1 \times$ NVIDIA A100 |
| # Training samples | 126,689/1,281,167 |
| # Evaluating samples | 5,000/50,000 |
| Optimizer | Adam |
| $\beta_1, \beta_2$ | 0.9, 0.999 |
| Weight decay | $5 \times 10^{-5}$ |
| Max learning rate (lr) | $1 \times 10^{-3}$ |
| Min learning rate (lr) | $1 \times 10^{-5}$ |
| LR decay | Cosine |
| Warmup epochs | 5 |
| Input size | 224 |

Table 12: Model configurations (Small Scaled-up) for image classification on ImageNet-100/1K.

| Configurations | Value |
|---|---|
| Patch size | 16 |
| Layer | 1 |
| Iterations $L$ | 12 |
| Hidden dimension $d$ | 512 |
| Feedforward ratio $M$ | $d$ |
| Number of heads $H$ | 8 |
| Positional encoding | Sinusoidal |
| Time embedding condition | $X_t$ |
| Time embedding frequency | 512 |

Table 13: Model configurations of different sizes.

| Configurations | Small | Small Scale-up | Base |
|---|---|---|---|
| Hidden dimension $d$ | 384 | 512 | 768 |
| Number of heads $H$ | 6 | 8 | 12 |

chosen between $[0, 0.4]$, and the masked region is replaced by a learnable token. Training loss is computed only for the masked tokens. We also follow the iterative decoding process in Chang et al. (2022) with temperature $= 1$ and decoding step $T = 24$. We also remove the MLP following the time embedding and set the embedding frequency equal to the hidden dimension to save parameters, and we find out that this implementation works better. Tables 14 and 15 show the detailed training recipe and configurations.

We evaluate the quality of the reconstructed images of masking out $40\%$ of the images with Base configurations. We report Peak Signal-to-Noise Ratio (PSNR), Structural Similarity Index Measure (SSIM) (Wang et al., 2004), Multi-Scale SSIM (MS-SSIM) (Wang et al., 2003), Learned Perceptual Image Patch Similarity (LPIPS) (Zhang et al., 2018) and Fréchet Inception Distance (FID) (Heusel et al., 2017) on the validation set (5k).

Table 14: Training recipe for masked image modeling.

| Hyperparameter | Value |
|---|---|
| Epochs | 300 |
| Batch size | 256 (64 per GPU $\times$ 4) |
| # GPU | $4 \times$ NVIDIA A100 |
| # Training samples | 126,689 |
| # Evaluating samples | 5,000 |
| Optimizer | AdamW |
| $\beta_1, \beta_2$ | 0.9, 0.96 |
| Weight decay | 0.1 |
| Learning rate (lr) | $1 \times 10^{-4}$ |
| LR decay | None |
| Gradient clipping | 3.0 |
| Input size | 256 |

Table 15: Model configurations for masked image modeling.

| Configuration | Value |
|---|---|
| Vocabulary size | 1,025 |
| Layer | 1 |
| Iterations | 12 |
| Hidden dimension $d$ | 768 |
| Feedforward ratio $M$ | $d$ |
| Number of heads $H$ | 12 |
| Positional encoding | Sinusoidal |
| Time embedding condition | $X_0$ |
| Time embedding frequency | 768 |
| Total parameters | 3.94 M |

## D.5 DEFINITION OF EFFECTIVE RANK AND AVERAGE ANGLE

We provide the formal definition of effective rank Eq. 17 and average angle Eq. 18 below. The effective rank is a continuous proxy of the full rank and, similar to the average angle, reflects the extent to which a set of vectors distributes uniformly.

**Definition 1** (Effective Rank). *For a matrix $\boldsymbol{X} \in \mathbb{R}^{d \times N}$, let $\Sigma = [\sigma_1, \ldots, \sigma_r]$ be its singular values where $r$ is its full rank and denote $p_i = \sigma_i / \sum_{j=1}^{r} \sigma_j$ the discrete probability. The effective rank (Roy & Vetterli, 2007; Guo et al., 2023) is defined as the exponential of the entropy*

$$\exp(-\sum_{i=1}^{r} p_i \log p_i). \tag{17}$$

**Definition 2** (Average Angle). *Given a set of vectors $\boldsymbol{X} = [\boldsymbol{x}_1, \ldots, \boldsymbol{x}_N] \in \mathbb{R}^{d \times N}$, the average angle of these vectors is*

$$\arccos \frac{2}{N(N-1)} \sum_{i=1}^{N} \sum_{j=i+1}^{N} \frac{\boldsymbol{x}_i^\top \boldsymbol{x}_j}{\|\boldsymbol{x}_i\|_2 \|\boldsymbol{x}_j\|_2}. \tag{18}$$

## D.6 COMPARISONS OF PARAMETER EFFICIENCY AND COMPUTATIONAL COST

Table 16: Comparisons of parameters and computational cost of different architectures.

| Models | # Params (M) | Memory (MB) | GFLOPs | Runtime (ms) |
|---|---|---|---|---|
| Transformer | 2.38 | 528 | 12.27 | $4.99_{\pm 0.14}$ |
| CRATE-T (Hu et al., 2024c) | 0.91 | 528 | 5.90 | $4.61_{\pm 0.17}$ |
| CRATE (Yu et al., 2023) | 1.06 | 528 | 5.90 | $4.86_{\pm 0.19}$ |
| Energy Transformer (Hoover et al., 2024) | 1.50 | 670 | 6.19 | $21.95_{\pm 0.35}$ |
| HYPER-SET (Ours) | 1.55 | 528 | 6.81 | $8.03_{\pm 0.21}$ |

Table 17: Impact of step size and normalization strategies on computational cost and runtime. The configuration of our HYPER-SET is highlighted in gray, which is designed to have RMSNorm placed after the subspace projection. Importantly, this configuration is not independent or disentangled but rather derived based on the proposed principled objective.

| Step Size Strategy | Pre-Norm | LayerNorm | # Params (M) | GFLOPs | Runtime (ms) |
|---|---|---|---|---|---|
| | | | 1.55 | 6.81 | $8.03_{\pm 0.21}$ |
| Learnable | ✓ | | 1.55 | 6.81 | $8.05_{\pm 0.22}$ |
| | | ✓ | 1.55 | 6.84 | $7.31_{\pm 0.26}$ |
| | ✓ | ✓ | 1.55 | 6.82 | $7.21_{\pm 0.28}$ |
| | | | 0.91 | 4.98 | $5.70_{\pm 0.15}$ |
| Fixed | ✓ | | 0.91 | 4.98 | $5.53_{\pm 0.25}$ |
| | | ✓ | 0.91 | 5.01 | $4.86_{\pm 0.21}$ |
| | ✓ | ✓ | 0.91 | 5.00 | $4.74_{\pm 0.18}$ |

To show the differences in parameters and computational cost incurred only by distinct architectures, we provide comparisons under the same width $d = 384$. Empirically, we compare FLOPs and runtime in one forward pass using `calflops`, measured on a single NVIDIA A100 GPU, of our model and other baselines with a $3 \times 224 \times 224$ random input. Table 16 reports the results with mean and standard deviations averaged over 1,000 runs for robust measurements. Our model has fewer FLOPs as it has inherent structures like weight sharing due to mathematical design.

To better understand the runtime performance of our model, we break down the contribution of key components in Table 17. The results reveal that the primary bottleneck in execution time stems largely from the facts that:

- We use an additional network to learn step sizes instead of keeping them fixed ($\sim 2.33$ ms);
- The current default implementation of `torch.nn.RMSNorm()`, which we use to meet the hyperspherical constraints, is slower than `torch.nn.LayerNorm()` ($\sim 0.72$ ms).

Currently, we trade some efficiency to allow for strong performance while keeping the principled design as transparent as possible with the additional modulation network for learned step sizes, which is adopted only for performance reasons rather than being a theoretically necessary component in our framework. In fact, our ablations that remove this extra network and fix the step sizes give similar runtime (5.70 ms in Table 17), but the performance decreases greatly (81.45 on CIFAR-10 and 58.29 on CIFAR-100 in Table 4). So, we believe there is still room for improvement in optimizing the step sizes schedule and further enhancing this minimalist implementation in future work.

## E    ADDITIONAL RESULTS OF MASKED IMAGE MODELING

Table 18 summarizes the results of masked image modeling with different masking ratios. When scaled to larger iterations and a wider feedforward module, our model achieves comparable results to Transformer but still slightly lags behind. This suggests the scalability of our model to large configurations may be a bottleneck for its development and deployment. More visual comparisons are provided in Figure 8.

Table 18: Comparisons of masked image modeling performance of varied masking ratios.

| Masking Ratio | Model | Iterations / FF Ratio $M$ | # Params (M) | PSNR ($\uparrow$) | SSIM ($\uparrow$) | MS-SSIM ($\uparrow$) | LPIPS ($\downarrow$) | FID ($\downarrow$) |
|---|---|---|---|---|---|---|---|---|
| 10% | Transformer | 12 / $4d$ | 8.85 | **17.693** | **0.466** | **0.709** | 0.236 | **22.428** |
| | HYPER-SET (Ours) | 12 / $d$ | 3.94 | 17.553 | 0.462 | 0.701 | 0.243 | 24.665 |
| | HYPER-SET (Ours) | 24 / $8d$ | 8.07 | 17.673 | 0.465 | 0.708 | **0.236** | 22.517 |
| 20% | Transformer | 12 / $4d$ | 8.85 | **17.185** | **0.451** | **0.678** | **0.261** | **27.320** |
| | HYPER-SET (Ours) | 12 / $d$ | 3.94 | 16.988 | 0.444 | 0.662 | 0.275 | 33.637 |
| | HYPER-SET (Ours) | 24 / $8d$ | 8.07 | 17.170 | 0.450 | 0.676 | 0.262 | 28.120 |
| 30% | Transformer | 12 / $4d$ | 8.85 | **16.616** | **0.435** | **0.642** | **0.291** | **35.095** |
| | HYPER-SET (Ours) | 12 / $d$ | 3.94 | 16.365 | 0.427 | 0.621 | 0.314 | 45.642 |
| | HYPER-SET (Ours) | 24 / $8d$ | 8.07 | 16.590 | 0.434 | 0.638 | 0.294 | 35.128 |

## F    RANK AND AVERAGE ANGLE OF EACH HEAD

### F.1    SUDOKU DATASET

Figures 9a and 9b capture the evolution of the effective rank and average angle of all heads. Most of them follow the separation dynamics on the hypersphere where tokens tend to be near-orthogonal, corroborating our design goal of attention energy.

### F.2    CIFAR-10 DATASET

The full results on CIFAR-10 also possess similar trends to those on the Sudoku dataset, as shown in Figures 10a and 10b.

### F.3    COMPARISONS WITH TRANSFORMER WITH SHARED QUERY, KEY, AND VALUE MATRIX

A notable connection between HYPER-SET and vanilla Transformer lies in the shared query (Q), key (K), and value (V) projection matrix, which has been studied recently (Kowsher et al., 2025). To verify whether HYPER-SET captures essential Transformer behaviors, we adapt vanilla Transformer to have shared QKV projections and measure its effective rank and average angle among projected tokens. Furthermore, we also include comparisons with HYPER-SET that adopts fixed step sizes set as 0.1 to evaluate the necessity of learned ones.

As shown in Figures 11a and 11b, both HYPER-SET and its fixed-step variant exhibit increasing token separation across subspaces, confirming the emergence of distributional uniformity and the benefit of learned step sizes. This dynamics is also mirrored in shared QKV Transformer, which cross-validates our insights on distributional uniformity in subspaces, suggesting the promise of this parameter-sharing design. In contrast, modifying the shared Transformer to reverse the update direction of attention—similar to the design in Eq. 9—leads to a decline in both rank and angle, highlighting a breakdown in uniformity. This contrast emphasizes that HYPER-SET is not a heuristic tweak of vanilla Transformer but a principled architecture.

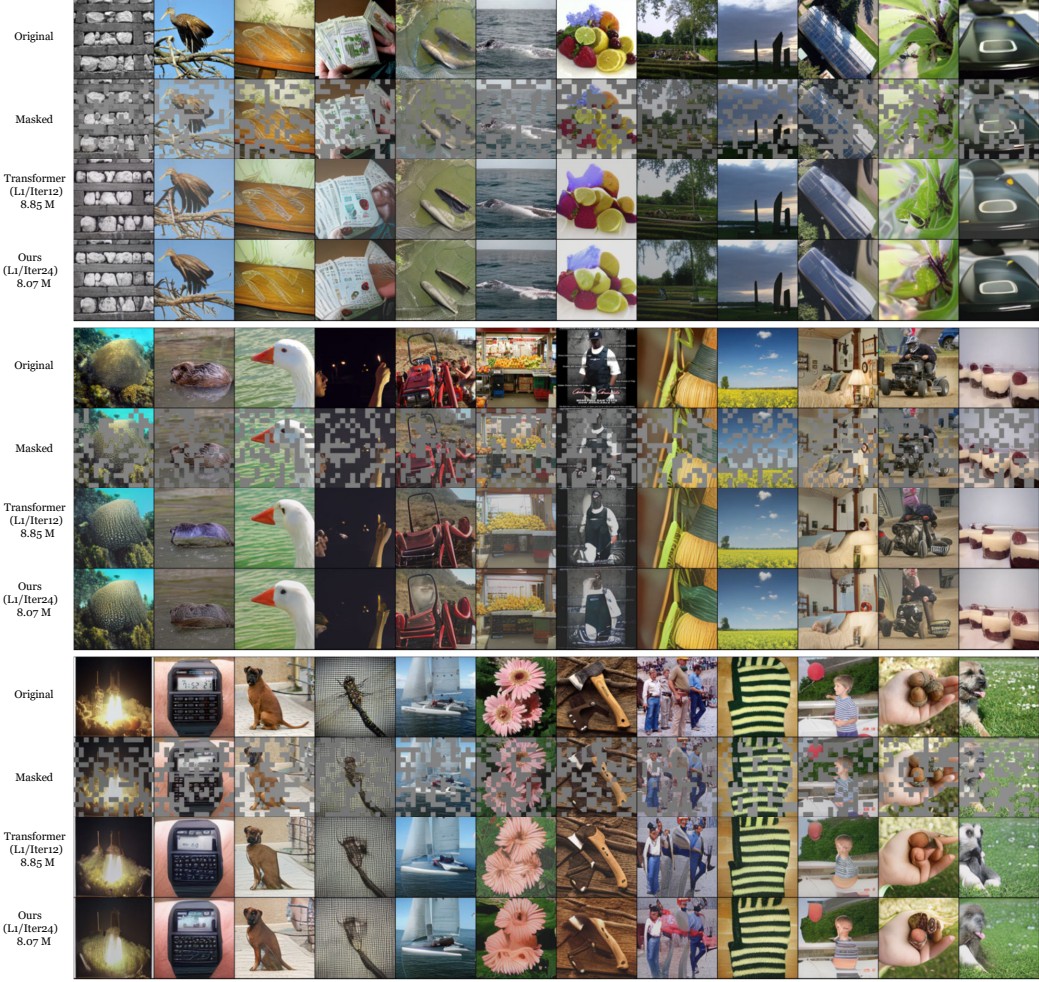

Figure 8: Visual comparisons of masked image modeling on ImageNet 256×256. Our model, when increased to Transformer scale with additional compute, can achieve similar reconstruction quality with 40% masking ratio.

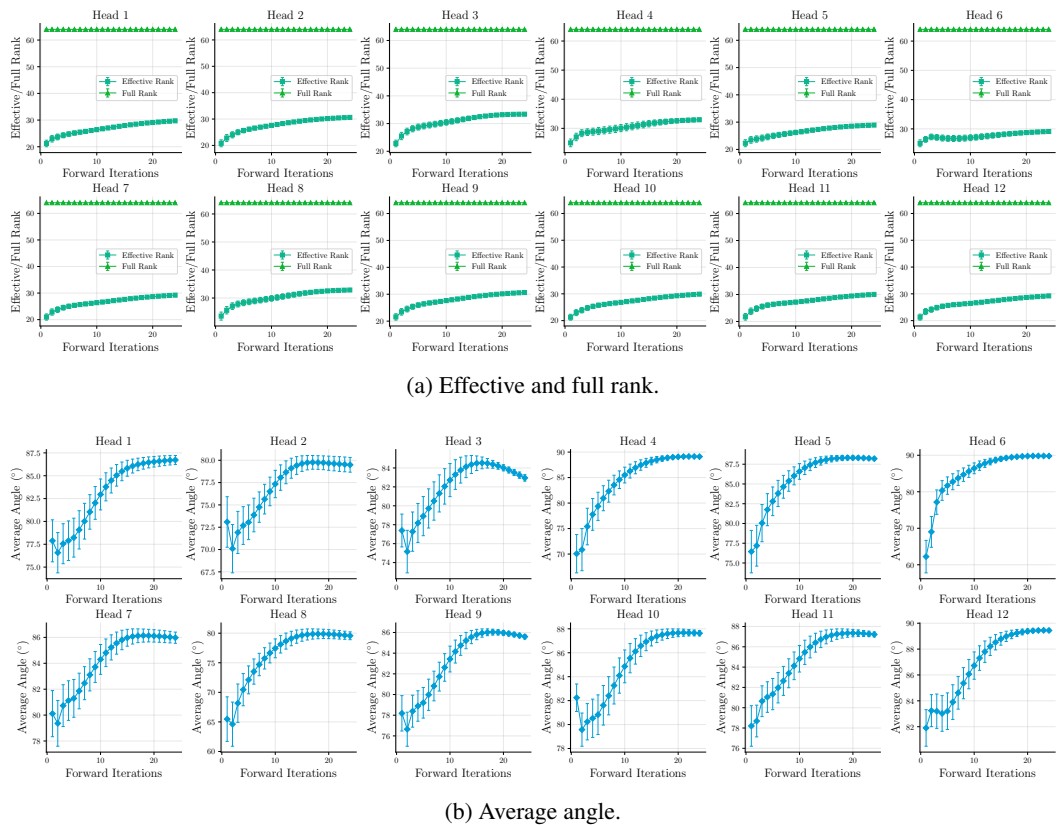

Figure 9: Geometric analysis on Sudoku. The effective rank (a) and average angle (b) of tokens projected to each subspace on the test set (Palm et al., 2018).

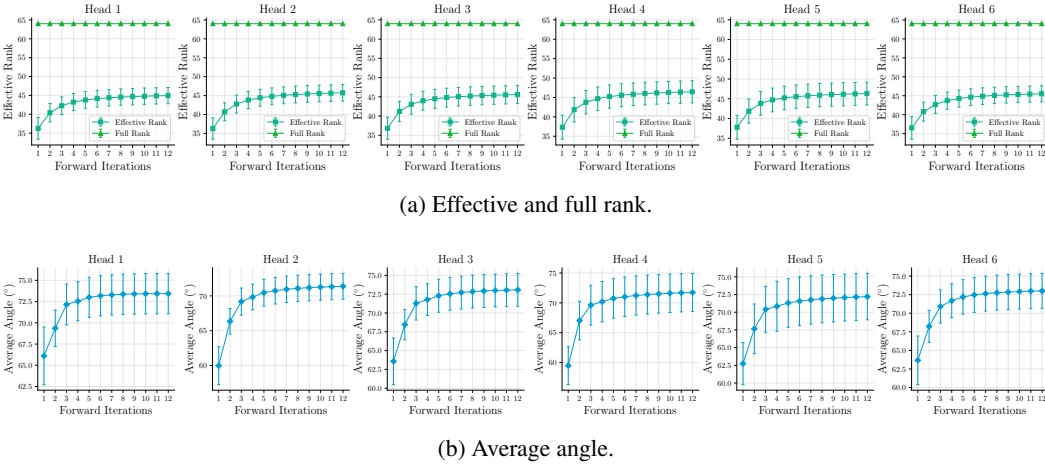

Figure 10: Geometric analysis on CIFAR-10. The effective rank (a) and average angle (b) of tokens projected to each subspace on the test set (Palm et al., 2018).

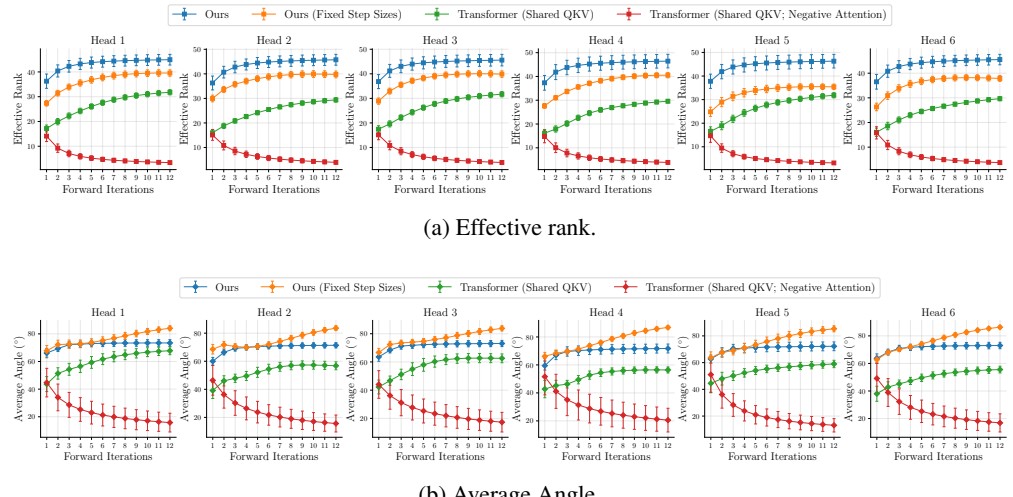

(a) Effective rank.

(b) Average Angle.

Figure 11: Impact of weight sharing and update rules. This layer-wise evolution of effective rank (*Top*) and average angle (*Bottom*) on the CIFAR-10 validation set verifies that our architecture and the variant using fixed step sizes share a similar functional role with Transformer with shared Query/Key/Value (QKV) matrices.

## G  ADDITIONAL RESULTS ON ALTERNATIVE DESIGNS AND SCALABILITY

### G.1  ALTERNATIVE DESIGNS ON ENERGY FUNCTIONS

Our proposed energy functions $E_{\text{ATTN}}$ and $E_{\text{FF}}$ provide an avenue to quantify the objective on uniformity and alignment in an amenable way for optimization. To manifest the significance of our conceptualization in designing a variety of Transformer-based models, we extend the energy functions to more general forms and provide alternative instantiations of them that can induce novel structures. Specifically, we generalize the energy functions in Eq. 3 and Eq. 5 to the following forms:

$$E_{\text{ATTN}} = \sum_{h=1}^{H} E_{\text{ATTN}}^h = \sum_{h=1}^{H} \sum_{i=1}^{N} f\left( \sum_{j=1}^{N} K(\boldsymbol{W}_h^\top \boldsymbol{x}_i, \boldsymbol{W}_h^\top \boldsymbol{x}_j) \right), \tag{19}$$

$$E_{\text{FF}} = -\sum_{i=1}^{N} g\left( \sum_{m=1}^{M} h(\boldsymbol{d}_m^\top \boldsymbol{x}_i) \right). \tag{20}$$

where $K : \mathbb{R}^p \times \mathbb{R}^p \to \mathbb{R}$ is a kernel function and $f, g, h : \mathbb{R} \to \mathbb{R}$ are non-decreasing scalar functions. For clarity, we omit the hyperspherical constraint but its correspondence to RMSNorm still holds.

By choosing different variations on these functions, we arrive at alternative energy functions and their consequent attention and feedforward operators, of which we summarize the specifications in the following tables. Remarkably, in Table 19, we can derive an attention with linear complexity $\mathcal{O}(N)$ by specifying the kernel function with the inner product of an element-wise transformation $\Phi : \mathbb{R} \to \mathbb{R}$, bridging our energy view with recent advances in linear attention design. In practice, we choose it as the sigmoid function $\sigma(x) = 1/(1 + \exp(-x))$, but other designs can also be possible. In Table 20, if we specify the outer function $g$ in the feedforward energy as a quadratic function, there is a novel summation and a Hadamard product operation emerging with the transformation $\Phi$, similar to the gating mechanism. We also specify $\Phi$ with the sigmoid function.

In summary, these design choices demonstrate that HYPER-SET is more than just a single model, but a blueprint for constructing principled Transformer variants. Each component—self-attention, feedforward, and normalization—can be systematically interpreted and designed within our energy minimization framework, providing a pathway for principled, modular innovation in sequence model architectures.

Table 19: Alternative designs on attention energy $E_{\text{ATTN}}$ and the induced operators.

| Operator | $f(x)$ | $K(\boldsymbol{x}, \boldsymbol{y})$ | $E_{\text{ATTN}}$ | $-\nabla_{\boldsymbol{x}} E_{\text{ATTN}}$ |
|---|---|---|---|---|
| Bi-Softmax (Default) | $\beta^{-1} \log(x)$ | $\exp(\beta \boldsymbol{x}^\top \boldsymbol{y})$ | Eq. 3 | Eq. 7 |
| Sigmoid Attention | $\frac{\beta^{-1}}{2} x$ | $\sigma(\beta \boldsymbol{x}^\top \boldsymbol{y})$ | $\frac{1}{2} \sum_{h=1}^{H} \sum_{i,j=1}^{N} \sigma\left(\beta (\boldsymbol{W}_h^\top \boldsymbol{x}_i)^\top \boldsymbol{W}_h^\top \boldsymbol{x}_j\right) \beta^{-1}$ | $\sum_{h=1}^{H} \boldsymbol{W}_h \boldsymbol{W}_h^\top \boldsymbol{X} \sigma(1-\sigma)\left(\beta(\boldsymbol{W}_h^\top \boldsymbol{X})^\top \boldsymbol{W}_h^\top \boldsymbol{X}\right)$ |
| Linear Attention | $\frac{\beta^{-1}}{2} x$ | $\frac{1}{2}\left(\beta \Phi(\boldsymbol{x})^\top \Phi(\boldsymbol{y})\right)^2$ | $\frac{1}{4} \sum_{h=1}^{H} \sum_{i,j=1}^{N} \left(\beta \Phi(\boldsymbol{W}_h^\top \boldsymbol{x}_i)^\top \Phi(\boldsymbol{W}_h^\top \boldsymbol{x}_j)\right)^2 \beta^{-1}$ | $\sum_{h=1}^{H} \boldsymbol{W}_h \Phi'(\boldsymbol{W}_h^\top \boldsymbol{X}) \odot \left(\beta \Phi(\boldsymbol{W}_h^\top \boldsymbol{X}) \Phi(\boldsymbol{W}_h^\top \boldsymbol{X})^\top \Phi(\boldsymbol{W}_h^\top \boldsymbol{X})\right)$ |

Table 20: Alternative designs on feedforward energy $E_{\text{FF}}$ and the induced operators.

| Operator | $g(x)$ | $h(x)$ | $E_{\text{FF}}$ | $-\nabla_{\boldsymbol{X}} E_{\text{FF}}$ |
|---|---|---|---|---|
| ReLU FF (Default) | $x$ | $\frac{1}{2} \text{ReLU}^2(x)$ | Eq. 5 | Eq. 10 |
| Softmax FF | $\log(x)$ | $\exp(x)$ | $-\sum_{i=1}^{N} \log\left(\sum_{m=1}^{M} \exp(\boldsymbol{d}_m^\top \boldsymbol{x}_i)\right)$ | $\boldsymbol{D} \underbrace{\text{softmax}}_{\text{column-wise}}(\boldsymbol{D}^\top \boldsymbol{X})$ |
| Gated FF | $\frac{1}{2} x^2$ | $\Phi(x)$ | $-\frac{1}{2} \sum_{i=1}^{N} \left(\sum_{m=1}^{M} \Phi\left(\boldsymbol{d}_m^\top \boldsymbol{x}_i\right)\right)^2$ | $\boldsymbol{D} \underbrace{\Phi(\boldsymbol{D}^\top \boldsymbol{X})}_{\text{column sum}} \odot \Phi'(\boldsymbol{D}^\top \boldsymbol{X})$ |

## G.2 PRELIMINARY RESULTS ON SCALABILITY

**Setups for Depth-wise Flexible Computation.** To equip our model with flexible computation while maintaining its core recurrence-based parameter sharing, we add an independent low-rank adaptation (Hu et al., 2022) matrix $W = AB$ to every iteration of the Transformer layer sharing the same base parameters. Inspired by but unlike depth-wise adaptation of pre-trained models in Bae et al. (2025), we train the base parameters and the adaptation matrix together. Both matrix $A \in \mathbb{R}^{d \times r}, B \in \mathbb{R}^{r \times d}$ with rank $r$ are initialized with Gaussain of 0.02 standard deviation. The scaling factor before the adaptation matrix is set as 4.

**Scaling Results on Images.** To further demonstrate the capability of our model when scaling up its size, we perform a preliminary evaluation on image classification. Following the experimental setups in the main paper to configure the models with one layer and repeat with 12 iterations, we scale up the width of Transformer to 1152 and ours to 1536, resulting in similar parameter size. The result is presented in Table 21. These explorations showcase the potential of HYPER-SET in improving model capacity without significantly increasing the total parameter count.

Table 21: Top-1 accuracy for image classification when scaling up the model size. Our model surpasses other baselines while being parameter-efficient compared to vanilla Transformer.

| Dataset | Model | Width $d$ | # Params (M) | Accuracy (%) |
|---|---|---|---|---|
| CIFAR-10 | Transformer | 1152 | 16.0 | 89.42 |
| | CRATE (Yu et al., 2023) | 1152 | 4.1 | 88.77 |
| | Energy Transformer (Hoover et al., 2024) | 1152 | 8.0 | 76.21 |
| | HYPER-SET (Ours) | 1536 | 14.3 | **90.62** |
| CIFAR-100 | Transformer | 1152 | 16.1 | 62.83 |
| | CRATE (Yu et al., 2023) | 1152 | 4.2 | 63.39 |
| | Energy Transformer (Hoover et al., 2024) | 1152 | 8.1 | 55.47 |
| | HYPER-SET (Ours) | 1536 | 14.4 | **66.30** |

**Scaling Results on Texts.** We also provide the results for text classifications to show the potential of HYPER-SET. We use `bert-base-uncased` from Hugging Face as the tokenizer with a maximum sequence length of 128. The number of recurrence is set to 6 (L1R6) and width is set to 384. The training recipe is the same as the image classification in Table 9 except we train for 10 epochs. The results on `yelp_review_full` dataset, a 5-way classification task, are shown in Table 22.

Table 22: Top-1 accuracy for text classification on `yelp_review_full` dataset.

| Model | Width $d$ | # Params (M) | Accuracy (%) |
|---|---|---|---|
| Transformer | 384 | 13.49 | 60.09 |
| CRATE-T (Hu et al., 2024c) | 384 | 12.02 | 59.43 |
| CRATE (Yu et al., 2023) | 384 | 12.17 | 59.18 |
| Energy Transformer (Hoover et al., 2024) | 384 | 12.61 | 54.93 |
| HYPER-SET (Ours) | 384 | 12.66 | **60.75** |

# H  DISCUSSION, LIMITATIONS AND FUTURE DIRECTIONS

## H.1  CONNECTIONS AND DIFFERENCES WITH ENERGY TRANSFORMER

Both our HYPER-SET and ET (Hoover et al., 2024) are grounded in the idea of interpreting Transformer components as gradient flows that minimize an energy function. However, the two approaches diverge significantly in motivations, theoretical formulation, and architectural design.

- **Motivation**
  - **HYPER-SET**: centers on a dual objective of semantic alignment (mode seeking) and uniformity (mass covering) under hyperspherical constraints grounded in maximal likelihood estimation. The proposed Hopfield-style energy aims to quantify it into an optimizable objective and serves as a specific instantiation of this more general principle, which fundamentally differs from ET.
  - **ET**: maintains the mechanistic interpretation of associative memory and does not directly connect the energy formulation to any particular representational challenge. Moreover, ET adopts Hopfield energy more as a starting point than as a motivation-driven design.

- **Methodology**
  - **HYPER-SET**: provides a more rigorous formulation of energy minimization. Our energy functions are defined *directly* on tokens Eq. 6 under a hyperspherical constraint. This formulation enables us to derive RMSNorm as a natural outcome of energy minimization in low-dimensional subspaces Eq. 8.
  - **ET**: defines energy over pre-normalized tokens (see Eq.(1)(6) in Hoover et al. (2024)) rather than tokens per se, bypassing the constrained optimization step. As a result, the role of normalization in ET is more heuristic than principled, following standard pre-norm practices rather than emerging from the underlying energy.

- **Implementations**
  - **HYPER-SET**: a) applies alternating minimization that results in attention and feedforward modules *sequentially*, reflecting the original Transformer structure; b) adopts adaptive, learnable step sizes conditioned on the input and iteration index Eq. 12, allowing the model to modulate its energy descent dynamically.
  - **ET**: a) performs energy updates via auto-differentiation that results in a *parallelized* fashion; b) uses fixed step sizes.

- **Empirical Verification**
  - **HYPER-SET**: a) confirms that the designed energy decreases in the forward pass in Figures 5 and 6; b) supports generalizations beyond softmax attention (e.g., different energy functions leading to alternative attention schemes), as illustrated in Tables 19 and 20.
  - **ET**: a) does not offer such explicit verification, leaving it unclear whether its dynamics faithfully track the energy descent objectives; b) Additionally, HYPER-SET achieves competitive performance with vanilla Transformers on tasks such as image classification and inference (e.g., Sudoku), whereas ET does not demonstrate comparable results in these domains.

## H.2  PRACTICAL IMPLICATIONS AND BROADER IMPACT

- **Why Study This Model**: This work proposes a principled approach to Transformer design by modeling representation learning as an energy minimization on the hyperspace. Unlike prior efforts such as Energy Transformer (Hoover et al., 2024) and CRATE (Yu et al., 2023), which

either diverge from their theoretical formulations or lack generality to derive new architectures, our model directly formulates energy functions on tokens with improved rigorousness, while supporting a spectrum of alternative designs.

- **Why CRATE Is a Fair Baseline**: CRATE (Yu et al., 2023) also pursues transparency and principled design, which shares a similar spirit with our goal, making it an appropriate baseline. While engineering-heavy ViTs may excel in benchmarks, they often involve significant redundancy. We aim to advance more compact, describable, and empirically competitive model design for next-generation architectures.

- **Interpretability**: We view the forward pass of HYPER-SET as a dynamical system. It features greater interpretability than vanilla Transformer in the sense that this dynamics is more readily describable and characterized by a meaningful quantity-the energy function-and grounded in well-established principles as maximum likelihood estimation. Beyond being merely conceptual, this dynamics is quantitatively verifiable, providing an interpretable and testable framework for understanding representation evolution.

- **A General Principle Beyond Canonical Hopfield Energy**: The Hopfield energy we employ in the main paper serves as one instantiation under the proposed general principle. Our formulation allows for broader energy-based designs—such as kernel-based alternatives to $\mathrm{logsumexp}$—enabling principled generalizations beyond standard attention mechanisms.

## H.3 LIMITATIONS

While HYPER-SET offers a principled and empirically competitive formulation for Transformer design, it also comes with several limitations that highlight directions for future work:

- First, the choice of subspace in our conceptualization is less explored. The choice of uniform prior on the hypersphere in assumption 2 could be too strong in practice. Overly enforcing uniformity may be restrictive in some tasks. Moreover, the number of subspaces $H$ and its dimension $p$ are chosen heuristically. Their relationship, if any, with the real data distribution remains unclear.

- Second, the modulation network to learn step sizes introduces complexity. Although we use a modulation network to learn step sizes adaptively, tuning the configurations of this network still requires considerable effort. In addition, it introduces more computational complexity despite that the overall architecture is more parameter-efficient.

- Third, our experiments on scalability are still preliminary. We confirm competitive and superior performance on less than 20 million parameters and prove depth-wise LoRA scaling effectiveness. However, extensions to truly large-scale settings—e.g., billion-level—have yet to be demonstrated.

## H.4 FUTURE DIRECTIONS

We provide several promising future directions:

1. **Autoregressive Modeling**: HYPER-SET currently lacks a causal structure, limiting its use in autoregressive sequence modeling. Adapting to GPT-style models with causal masking is an important future step.

2. **Flow Matching and ODE Connections**: The iterative updates in HYPER-SET resemble neural ODEs, suggesting potential connections to flow matching techniques that may unify Transformer-based models with generative modeling.

3. **Scalability and Adaptive Computation**: Our initial results with depth-wise LoRA are promising but preliminary. Future work could explore dynamic iteration depth, inspired by latent space reasoning (Geiping et al., 2025), sparsity (Tan et al., 2023), and mixture-of-experts (Csordás et al., 2024).

