# OpenReview forum: "Hyper-SET: Designing Transformers via Hyperspherical Energy Minimization"
_ICLR.cc/2026/Conference — ICLR 2026 Poster_

### Official Review · Reviewer_fUR7 · 2025-10-25

**Soundness:** 2
**Presentation:** 3
**Contribution:** 3
**Rating:** 4
**Confidence:** 4

**Summary:**

The authors propose Hyper-SET architecture, a recurrent-depth transformer-like sequence-to-sequence architecture. It is constructed as a discretization of a constrained gradient flow on a modified Hopfield energy function restricted to the product of spheres. This network is a single layer iterated an arbitrary number of times, with one layers' worth of parameters being shared across every application of the layer. The layer itself is a novel construction obtained via gradient descent-like steps on different parts of the aforementioned Hopfield-like energy. The authors demonstrate via experimental results that under this highly parameter-constrained setup, the model performs competitively against other architectures formed in the same way (i.e., single layer iterated arbitrarily many times) with different architectures for the layer, e.g., a standard transformer module, across a diverse set of problems.

**Strengths:**

- The mathematical derivation seems complete, correct, and interesting: while cooking up some objective whose optimization unrolling/discretization reproduces a transformer-like architecture has been done before [1, 2, 3], only a few previous works such as [3] attempt to give a holistic interpretation to the objective or optimization procedure.
- The new block is novel and shows a genuine attempt at developing a prescriptive theory to improve the current practice.
- The paper is written well; the connection to previous work and situation in the literature is clear; and the current approach is well-motivated and rigorously exposited.

[1]: Ramsauer, Hubert, et al. "Hopfield networks is all you need." arXiv preprint arXiv:2008.02217 (2020).

[2]: Yang, Yongyi, and David P. Wipf. "Transformers from an optimization perspective." Advances in Neural Information Processing Systems 35 (2022): 36958-36971.

[3]: Yu, Yaodong, et al. "White-box transformers via sparse rate reduction." Advances in Neural Information Processing Systems 36 (2023): 9422-9457.

**Weaknesses:**

Some meaningful weaknesses:
- By the authors' own admission, the proposed network "suits better [sic] under a resource-constrained setting as its inherent structural biases could limit its scaling on large datasets." This seems like a downside compared to the regular Transformer whose benefits over other architectures _only_ manifest at scale. With this in mind, maybe comparisons to some simpler RNNs/convnets may be meaningful?
- Comparisons against other single-layer recurrent-depth baselines require some more explanation and elaboration (beyond Appendix D). See "Questions" Q1 below.
- The single-layer recurrent-depth setting is interesting but impractical. A well-trained ViT with 12 (non-tied) layers can easily get > 90 on CIFAR 10 for instance (cf Table 1 where the high score is 90, whereas even the original ViT paper [1] boasts 99% in Table 6). Almost all practical use cases do not use this weight sharing. With this in mind, the empirical results currently do not demonstrate the practical efficiency of such networks. See "Questions" Q2 below.
- Relatedly, the baseline deep networks in Figure 4 seem under-tuned, e.g., the baseline 12-layer ViT is said to get 66% on CIFAR100 in Table 4 but [1] Table 6 shows that a 12 layer ViT can get > 90% on CIFAR100. See "Questions" Q2 below.

Some nits:
- It does not really make sense to write two separate ODEs to show that you optimize each part of the features (e.g. (7) and (10)), which would suggest a parallel-structure network. Instead it is more mathematically sound to have a combined ODE with both RHS terms summed together, and obtain the network architecture via operator splitting (or some similar technique).
- Table 2 empirical performances in the first two columns are close and seem within-noise, requiring 5 significant figures on the benchmark to distinguish the models; within this context, bolding is done slightly incorrectly, e.g., only bolding Hyper-SET numbers in the case of a tie (0.417 in the second column). On the other hand, the text referencing Table 2 is slightly unfair to Hyper-SET since it can emphasize parity in the first two columns instead of saying it lags in all metrics.

[1]: Dosovitskiy, Alexey. "An image is worth 16x16 words: Transformers for image recognition at scale." arXiv preprint arXiv:2010.11929 (2020).

**Questions:**

Q1: Can you elaborate a little on the surprising performance of the baselines in the experiments? Namely,
- Q1.1: For Figure 3a, do you have intuition why otherwise reasonably-successful architectures completely fail this task and get a 0?
- Q1.2: For Table 2, do you have intuition why the energy transformer performance drops by a large amount on IN1K?

Q2: Can you train some non-weight-shared networks with more parameters, e.g., using a largely validated pipeline such as the original ViT pipeline [1], to simultaneously establish fair comparisons w.r.t. deep network baselines, and show in some more detail how the proposed network can be made practically performant (in either the single-layer recurrent-depth or deep settings)?

[1]: Dosovitskiy, Alexey. "An image is worth 16x16 words: Transformers for image recognition at scale." arXiv preprint arXiv:2010.11929 (2020).

---

> ### Author Response · Authors · 2025-11-20
> **Response 1**
>
> We thank the reviewer for the thoughtful comments and insightful questions, which help us improve the clarity and quality of our work. **Please note we have updated the manuscript in blue color** and will refer to the new versions of tables and figures in the responses.
>
> **Weaknesses**
>
> > By the authors' own admission, the proposed network "suits better [sic] under a resource-constrained setting as its inherent structural biases could limit its scaling on large datasets." This seems like a downside compared to the regular Transformer whose benefits over other architectures only manifest at scale. With this in mind, maybe comparisons to some simpler RNNs/convnets may be meaningful?
>
> It needs to be noted that we make this argument under the **single-layer recurrent-depth** setting, which is the major focus of this work because of ease to analysis. In this setting, our model can outperform standard Transformer with similar size on small to medium-sized datasets (CIFAR-10, CIFAR-100, ImageNet-100) but slightly lags on large-scale ImageNet-1K (66.94 vs. 62.76) as shown in Table 1.
>
> In fact, with this single-recurrent-layer (L1R12) setting, it can even be quite difficult for CNNs that have strong inductive biases to optimize and perform well. To ensure the architectural difference plays the role, we compare results in Table 1 with a single-layer recurrent ResNet block from PyTorch's implementation under similar parameters on CIFAR-10, which is shown below. Our model significantly outperforms it (90.11 vs. 74.94) while having similar number of parameters.
>
> |Model |Config (hidden $d$)|#Param| Acc. (CIFAR-10)|Acc. (CIFAR-100)|
> |:--- | :---: | :---: | :---: | :---: |
> |Transformer|L1R12 (384)|2.18M|89.90|61.89|
> |CRATE-T|L1R12 (896)|2.55M|87.54|60.23|
> |CRATE|L1R12 (768)|2.58M|84.81|58.22|
> |Energy Transformer|L1R12 (512)|2.11M|76.39|50.60|
> |Ours |L1R12 (512)|2.12M |**90.11**|**63.41**|
> |Recurrent ResNet|L1R12 (320)|1.86M |74.94|1.63|
>
>
>
> As for a more practical **multi-layer non-recurrent** setting, we have updated the paper with Table 2 that shows our model can actually scale almost equally well as standard Transformer while being more parameter-efficient. This suggests our model can have the benefit of scaling in Transformer when extended practically, meanwhile being more useful in resource-constrained settings.

---

> ### Author Response · Authors · 2025-11-20
> **Response 2**
>
> **Weaknesses & Questions**
>
> > Comparisons against other single-layer recurrent-depth baselines require some more explanation and elaboration (beyond Appendix D). See "Questions" Q1 below.
> >
> > Q1: Can you elaborate a little on the surprising performance of the baselines in the experiments? Namely,
> >
> > Q1.1: For Figure 3a, do you have intuition why otherwise reasonably successful architectures completely fail this task and get a 0?
>
>
> We first want to point out that Sudoku may be hard for some architecture to achieve non-zero accuracy, because the 9x9 Sudoku grid has to be filled in with some exact digits to meet the constraints which may be hard to learn without proper inductive bias. As shown in Figure 1 of [1], even some foundation models have zero accuracy.
>
> As we state in Appendix H.2 about the broader impact of our paper, previous works diverge from their theoretical formulations, and the design objective are inspired by domain knowledge (e.g., information theory in CRATE [2]). These "white-box" models inevitably infuse some prior assumptions or biases into the architecture, but their formulation may not characterize the information processing in the right way, and the optimization-induced layer needs to be implemented extremely carefully, that would otherwise make the whole model more brittle than Transformer which has fewer biases.
>
> Therefore, a reasonable explanation would be that Hyper-SET succeeds because we have a better characterization or assumption about the design objective, which is based on as fundamental as maximum likelihood estimation and thus more realistic, and we rigorously follow our formulation without deviations in implementation, making the architecture better aligned with the optimization procedure. This could allow Hyper-SET to enjoy both the principled design and the expressivity of Transformer.
>
>
> > Q1.2: For Table 2, do you have intuition why the energy transformer performance drops by a large amount on IN1K?
>
> I assume the reviewer is referring to Table 1 in the original manuscript. We have provided a comprehensive comparison and discussion on the major differences from energy transformer in Appendix H.1 in terms of motivation, methodology, and implementation.
>
> To highlight, energy transformer only maintains the mechanistic interpretation of associative memory and does not directly connect to any particular representational challenges, which is crucial for general task performance like image classification. This domain-specific formulation greatly impedes its scalability and generality. In contrast, our model is derived from a principled formulation that directly addresses the representational challenges via maximum likelihood on the hypersphere, leading to a more effective architecture for general tasks.

---

> ### Author Response · Authors · 2025-11-20
> **Response 3**
>
> > Q2: Can you train some non-weight-shared networks with more parameters, e.g., using a largely validated pipeline such as the original ViT pipeline [1], to simultaneously establish fair comparisons w.r.t. deep network baselines, and show in some more detail how the proposed network can be made practically performant (in either the single-layer recurrent-depth or deep settings)?
>
> To showcase the extension to non-weight-shared setting,  we have originally provided in Figure 4 a comparison to standard Transformer on CIFAR-100 with different proportion of weight-sharing. We outperform a fully non-recurrent (12-layer in this case) Transformer that is considered to have an upper performance (gray dashed line).
>
> To further demonstrate the practicality of non-recurrent settings, which is more commonly used in large-scale applications, we have updated the paper with comparisons in a 12-layer non-recurrent setting on ImageNet-1K in Table 2, as mentioned before. The pretraining recipe on ImageNet-1K is the same as experiments Table 1 while fine-tuning is done by adjusting batch size to 256, epochs to 50, learning rate to 1e-4 and weight decay to 1e-5. We list the main results here.
>
> $*$ means first pretraining on ImageNet-1K (IN-1K) and then fine-tuning on CIFAR10/100.
>
> |Model |Config (width $d$)|#Param|Acc. (IN-1K)| Acc. (CIFAR10)|Acc. (CIFAR100)| Acc. (CIFAR10)*|Acc. (CIFAR100)*|
> |:--- | :---: | :---: | :---: | :---: |:---: |:---: |:---: |
> |Transformer|L12R1 (384)|21.86M|**67.90**|87.44|62.84|**96.95**|**83.10**|
> |CRATE|L12R1 (512)|10.28M|60.96|88.76|63.94|94.18|77.39|
> |Ours |L12R1 (512)|8.17M |66.26|**88.82**|**64.98**|95.76|80.89|
> |Ours |L12R1 (768)|17.56M |67.20|88.53|64.16|96.47|82.60|
>
>
>
> > It does not really make sense to write two separate ODEs to show that you optimize each part of the features (e.g. (7) and (10)), which would suggest a parallel-structure network. Instead it is more mathematically sound to have a combined ODE with both RHS terms summed together, and obtain the network architecture via operator splitting (or some similar technique).
>
> Please be note that we indeed optimize the combined energy in Eq.(6) by using the alternating minimization algorithm following [2] as mentioned in line 268, which is a kind of operator splitting method.
>
> We write the two ODEs separately in Eq.(7) and Eq.(10) just to provide more intuition on how each module works to optimize its corresponding part of the energy, which helps understand the functional role of each module.
>
>
> > Table 2 empirical performances in the first two columns are close and seem within-noise, requiring 5 significant figures on the benchmark to distinguish the models; within this context, bolding is done slightly incorrectly, e.g., only bolding Hyper-SET numbers in the case of a tie (0.417 in the second column). On the other hand, the text referencing Table 2 is slightly unfair to Hyper-SET since it can emphasize parity in the first two columns instead of saying it lags in all metrics.
>
> We have updated the results with 5 random seeds to reduce the variance and report the mean and standard deviation in Table 2 (now Table 3 in the new version). We also revised the bolding accordingly and modified the text to emphasize parity in the first two columns.
>
> ---
> [1] Wang et al, Hierarchical Reasoning Model, arXiv, 2025
>
> [2] Yu et al. White-box transformers via sparse rate reduction. NeurIPS, 2023.

---

> > ### Comment · Reviewer_fUR7 · 2025-11-23
> > **Reply to Authors**
> >
> > Thanks for the very detailed rebuttal. I have some remaining comments, which are as follows:
> >
> > > It needs to be noted that we make this argument under the single-layer recurrent-depth setting...
> >
> > Thanks for clarifying that the paper's claims about performance under resource constraints only apply to this setting. Furthermore, the results about comparing the new block to different blocks from RNN/CNN are quite reasonable and make the work better.
> >
> > > As for a more practical multi-layer non-recurrent setting, we have updated the paper with Table 2 that shows our model can actually scale almost equally well as standard Transformer while being more parameter-efficient.
> >
> > Thanks for completing these experiments in Table 2 of the paper. Unfortunately in this more common setting, at the present scale of 10-20M parameter models, the scalability of transformers and transformer-like architectures have not been demonstrated; for instance [1, Figure 4] shows that transformers only improve the performance at several hundred parameters and 10s/100s of millions of samples, compared to ResNet baselines. If this is all that can be demonstrated with the authors' resources, the result may still be interesting and merit inclusion, but the claim that the new architecture scales almost as well as the transformer is way too strong. Regardless, another fair comparison in the most common setting is a useful and worthwhile contribution.
> >
> > > Therefore, a reasonable explanation would be that Hyper-SET succeeds because...
> >
> > This is a strong claim about the relative superiority of the presented formulation and how this concretely translates to empirical results. Proving this would definitely merit more experiments, especially because the authors use some heuristics to actually design the network operators which optimize their objective. (Note that the other compared architectures Energy Transformer and CRATE also do this.)
> >
> > > To further demonstrate the practicality of non-recurrent settings... (IN1K pre-training)
> >
> > This experiment is great, I appreciate it especially because it gives a clearer picture of how the architecture performs in practical settings and at larger (data) scale. The transfer learning results are quite reasonable.
> >
> > > Please be note that we indeed optimize the combined energy in Eq.(6) by using...
> >
> > Yes, I believe I understand the idea behind the formulation and the realization via an operator-splitting-like step. The point is writing down the two subsequent split ODEs is not very rigorous since those equations do not _both_ govern the update of $\boldsymbol{X}$. I was just wondering whether the presentation around this point can be improved.
> >
> > Overall I think due to the revisions the paper has become much more solid. The real missing pieces, in my opinion, are:
> >
> > (1) the fair comparison against larger models (e.g. ViT-B to ViT-H scale), which allows a true analysis or study of the scaling characteristics;
> >
> > (2) a more complete empirical/theoretical understanding of the objective, and especially how the operators create representations which are useful for downstream tasks (especially those which are difficult for other models such as Sudoku).
> >
> > But because of this improvement and because many of my questions/weaknesses were addressed, I will increase the score.
> >
> > [1]: Dosovitskiy, Alexey. "An image is worth 16x16 words: Transformers for image recognition at scale." arXiv preprint arXiv:2010.11929 (2020).

---

> > > ### Author Response · Authors · 2025-11-28
> > >
> > > We are happy to see the reviewer's concerns have been addressed and grateful for further raising the score.
> > >
> > > We acknowledge that compared to the JFT pretraining of over 10M samples in [1], our Hyper-SET has yet to be tested on such a large scale; but our goal is mainly about demonstrating the possibility and practicality of principled design of a Transformer-like model, and it indeed shows potential in practical scenarios. We have softened the wording from "...scales almost as well as the transformer..." to "...potentially preserves the benefit of scaling in Transformers when extended practically..." in line 429. We will open-source the code upon publication for further investigation in the community.
> > >
> > > This reviewer's opinion on understanding of the objective and how to characterize what is a good representation for downstream tasks is indeed a deep and challenging open question. Our paper is a step towards these questions under the context of principled Transformer design. We hope it can be an invitation for more researchers to reveal the connections between Transformer-like architectures and representation learning.
> > >
> > > We again thank you for your time and the kind suggestions to improve the paper.
> > >
> > > ---
> > > [1]: Dosovitskiy, Alexey. "An image is worth 16x16 words: Transformers for image recognition at scale." arXiv preprint arXiv:2010.11929 (2020).

---

### Official Review · Reviewer_Dhku · 2025-10-31

**Soundness:** 4
**Presentation:** 4
**Contribution:** 3
**Rating:** 8
**Confidence:** 5

**Summary:**

This paper presents a principled, "top-down" derivation of components of the Transformer architecture from an energy-based learning perspective. The authors posit that effective token representations should simultaneously achieve two goals: semantic alignment (clustering of similar concepts) and distributional uniformity (spreading representations out to avoid collapse). They formulate these two objectives as a single, combined hyperspherical energy function. The paper then demonstrates that an iterative optimization process to minimize this energy function naturally yields the core components of a Transformer block: symmetric self-attention emerges from the uniformity objective, a symmetric feedforward network from the alignment objective, and RMSNorm from the hyperspherical constraint. The resulting model, HYPER-SET, is a recurrent, parameter-shared architecture. The authors evaluate HYPER-SET on tasks like Sudoku solving, image classification, and masked image modeling, comparing its performance and parameter efficiency to standard Transformers.

**Strengths:**

The paper is very well-written. The argument is presented with great clarity, smoothly taking readers from a conceptualization to mathematical derivations and finally to empirical validation.
The core premise of the work is very interesting. It is not the traditional heuristic-driven design of modern architectures; the "first principles" approach is compelling. The conceptualization of token dynamics as a balance between "semantic alignment" and "distributional uniformity" is intuitive and provides a compelling theoretical foundation for the architecture.
The mathematical derivations that connect these two principles to the standard Transformer components are elegant. Showing how attention, FFNs, and RMSNorm emerge as natural solutions to a single constrained energy minimization problem is the central theoretical strength of this work.
The empirical evaluation is extensive and respectable. The authors are transparent about the model's performance, showcasing its strengths in parameter efficiency and extrapolation on tasks like Sudoku, while also honestly reporting its limitations on larger-scale datasets. This thoroughness provides a complete picture of the model's utility and grounds the compelling theoretical framework in practical results.

**Weaknesses:**

The primary drawback, which the authors acknowledge through their experiments, is the model's difficulty to scale to larger datasets (like ImageNet-1K) and more complex generative tasks. On these, it lags behind standard Transformer baselines. While the authors are not expected to conduct a full-scale SOTA-level experiment, and their attempt to scale by stacking two distinct layers is a valuable inclusion, the paper would be significantly strengthened by a more in-depth discussion on how this scaling limitation could be overcome. The paper falls short in hypothesizing why this limitation exists. Is the hard parameter-sharing of the recurrent model creating an optimization bottleneck that a deep stack of independent layers avoids? The work would be more complete if it drew parallels to existing literature on deep vs. recurrent models or the challenges of weight-sharing in large-scale settings. A discussion on potential paths forward would provide a more comprehensive outlook. Without this, the proposed model, while theoretically compelling, feels disconnected from the path toward practical, large-scale application.

**Questions:**

For Sudoku extrapolation (Figure 3b), the authors attribute the gains to the proposed energy minimization dynamics. However, these improvements could maybe come from implicit regularization introduced by the symmetric parameter sharing and hyperspherical constraints, instead of the iterative energy optimization itself? Did the authors conduct ablations to disentangle the effects of the energy-based formulation from those of architectural regularization?

---

> ### Author Response · Authors · 2025-11-20
> **Response 1**
>
> We are extremely grateful for the acknowledgement from the reviewer and the kind suggestions to strengthen the paper. **Please note we have updated the manuscript in blue color** and will refer to the new versions of tables and figures in the responses.
>
> **Weakness**
> > While the authors are not expected to conduct a full-scale SOTA-level experiment, and their attempt to scale by stacking two distinct layers is a valuable inclusion, the paper would be significantly strengthened by a more in-depth discussion on how this scaling limitation could be overcome.
>
> > A discussion on potential paths forward would provide a more comprehensive outlook. Without this, the proposed model, while theoretically compelling, feels disconnected from the path toward practical, large-scale application.
>
> Thank you for the kind suggestions. We have included in our original submission a discussion on the limitations and future work in Appendix H.3 and H.4 and referrer to them in the conclusion, where we provide some suggestions on how to practically scale up the recurrent-depth model.
>
> To elaborate more, one potential path forward is to use LoRA-like techniques by adding a lightweight low-rank matrix to the base weight of the recurrent layer at each depth iterations. In this way, each layer can have a slightly different parameterization to increase the flexibility without a hard weight-sharing or significantly increasing the number of parameters. We have explored this possibility in Table 5.
>
> Another direction is to explore mixture-of-experts structures. In our current framework, the weight in the feedforward module can be considered as a set of basis vectors that define semantic directions to which each token should align. We can extend this idea to have multiple sets of basis vectors by redefining the corresponding energy functions $E_{\text{FF}}$. This would allow each token to align semantic directions combinatorially beyond $d$ vectors in one expert, which could increase the model's capacity without an increase in activated parameters.
>
> Finally, to demonstrate the practicality of non-recurrent settings, which is more commonly used in large-scale applications, we have updated the paper with additional comparisons in a 12-layer non-recurrent setting in Table 2, showing competitive performance with standard Transformers while being more parameter-efficient. This reduction in parameters can be consistently enlarged by stacking more distinct layers as shown in Figure 4 thanks to our parsimonious design, further underscoring its potential in practical large-scale scenarios.
>
> > The paper falls short in hypothesizing why this limitation exists. Is the hard parameter-sharing of the recurrent model creating an optimization bottleneck that a deep stack of independent layers avoids? The work would be more complete if it drew parallels to existing literature on deep vs. recurrent models or the challenges of weight-sharing in large-scale settings.
>
> Our take on this is that hard parameter sharing forces the model to find a fixed point of a single dynamical system. While this is excellent for reasoning tasks (like Sudoku) which require iterative refinement of the same logic, it struggles with general-domain perception tasks (like ImageNet) that require transforming low-level pixels into high-level concepts, presumably a hierarchical process. The shared weights must compromise to satisfy the requirements of all processing stages simultaneously, potentially creating the optimization bottleneck like the reviewer assumes. This could be the reason why recurrent models have yet to be proven universally beneficial in large-scale settings, other than in specific domains like reasoning tasks (as we make some parallels on test-time compute in reasoning in line 359).
>
> We also update the paper with an additional related work section on recurrence in Transformers to position our work within the broader context of existing literature.

---

> ### Author Response · Authors · 2025-11-20
> **Response 2**
>
> **Questions**
>
> > For Sudoku extrapolation (Figure 3b), the authors attribute the gains to the proposed energy minimization dynamics. However, these improvements could maybe come from implicit regularization introduced by the symmetric parameter sharing and hyperspherical constraints, instead of the iterative energy optimization itself? Did the authors conduct ablations to disentangle the effects of the energy-based formulation from those of architectural regularization?
>
> The architectural biases (e.g., symmetric weight, the position of hyperspherical constraint) are an integral part and direct derivation of our energy-based formulation on the hypersphere through unrolling. These components are not heuristically designed but based on certain functional roles with statistical and geometrical meaning (a.k.a, first-principled design), which is the core argument and contribution of this work. As such, **these components are not mutually independent or disentangled**, and it would be inappropriate and unfair to attribute the success on Sudoku to a specific one through ablations.
>
> To provide more intuition, we believe a reasonable explanation would be that Hyper-SET succeeds because we find a suitable way to model the design objective that characterizes the information processing of Transformer, which is based on as fundamental as maximum likelihood estimation and thus more realistic, and we rigorously follow our formulation without deviations in implementation, making the architecture better aligned with the energy optimization procedure. This could allow Hyper-SET to both enjoy the principled design and maintain the expressivity of Transformer (in fact, better in the Sudoku case).

---

### Official Review · Reviewer_jreb · 2025-11-01

**Soundness:** 3
**Presentation:** 3
**Contribution:** 2
**Rating:** 4
**Confidence:** 3

**Summary:**

The authors introduces HYPER-SET, a principled that reframes the Transformer as an iterative energy minimizer on a hypersphere. This hyperspherical energy is designed to balance two key goals: semantic alignment with important features and distributional uniformity to prevent representational collapse. By deriving the math for this optimization, core components like symmetric attention, feedforward layers, and RMSNorm emerge naturally rather than hand-designed. The final recurrent-depth model is parameter-efficient and demonstrates competitive performance on diverse tasks, including Sudoku solving, image classification, and masked image modeling.

**Strengths:**

1. The primary strength of this work is its "white-box" design. The authors start from a unified principle (hyperspherical energy minimization) and derive the architectural components (Bi-Softmax attention, FFN, RMSNorm) as the mathematical solution.

2. Figure 5 demonstrates that the designed energy function decreases during the forward pass. Figure 6 shows that the effective rank and average angle of tokens increase, empirically confirming that the "distributional uniformity" objective is being met. This link between theory and empirical dynamics is therefore good.

3. HYPER-SET outperforms other models in the "white-box" design class (CRATE and Energy Transformer).

4. Table 3 provides an ablation that validates the design choices. It shows that the components derived from the theory outperform more common alternatives.

**Weaknesses:**

W1. The model’s good results on iterative reasoning tasks (like Sudoku) do not generalize to general-domain tasks. On standard benchmarks like ImageNet-1K and masked image modeling, the vanilla Transformer baseline remains superior.

W2. The recurrent-depth design is a major practical drawback. As confirmed by the authors' runtime analysis in Table 15, the 1-layer, 12-iteration HYPER-SET model is significantly slower than a standard Transformer, despite having fewer parameters. This high latency severely limits its practicality.

**Questions:**

Q1. The model is slower than a standard Transformer (Table 15) and underperforms on general-domain tasks (Weakness 1). Given these latency and performance gaps, how do the authors justify the practical value of its parameter efficiency?

Q2. Does the training memory cost scale linearly with the number of iterations to store activations for backpropagation?

Q3. The paper's most significant performance gap appears on the Sudoku task, where both CRATE and Energy Transformer completely fail on this task. What is the authors' hypothesis for why these other "white-box" models fail catastrophically on this reasoning task, while HYPER-SET succeeds?

---

> ### Author Response · Authors · 2025-11-20
> **Response 1**
>
> We sincerely thank the reviewer for the thoughtful comments and constructive questions, which help us improve the clarity and quality of our work. **Please note we have updated the manuscript in blue color** and will refer to the new versions of tables and figures in the responses.
>
> **W1 & Q1**
>
> > The model’s good results on iterative reasoning tasks (like Sudoku) do not generalize to general-domain tasks. On standard benchmarks like ImageNet-1K and masked image modeling, the vanilla Transformer baseline remains superior.
> >
> > The model is slower than a standard Transformer (Table 15) and underperforms on general-domain tasks (Weakness 1). Given these latency and performance gaps, how do the authors justify the practical value of its parameter efficiency?
>
> We respectfully point out that in Sudoku and small to medium-sized datasets (CIFAR-10, CIFAR-100, ImageNet-100), our one-layer recurrent model can outperform Transformer (Table 1) with similar size, but indeed slightly underperform Transformer on large-scale ImageNet-1K (66.94 vs. 62.76, Table 1) **in the recurrent setting**.
>
> We have updated Table 15 (now Table 16) with a more accurate profiling of runtime by not storing the gradient. The updated results show that our layer is around 60% slower than Transformer layer (8.03ms vs. 4.99ms) **but with significantly fewer GFLOPs** (6.81 vs. 12.27).
>
> We want to highlight that parameter efficiency coming from structural biases, such as weight sharing, symmetric weight, innately arises as the direct result of principled design via unrolling the energy function. It suggests some part of the parameters can be reused for a parsimonious architecture; and we argue it perhaps works better under tasks with structures or resource-constrained setting (e.g., edge devices), where model size matters, rather than data/compute-intensive scenarios (line 410, Section 5.2).
>
> Another practical value of this one-layer parameter-efficiency is that it can be easily extended to multi-layer non-recurrent settings, where this reduction in parameters can be consistently enlarged by stacking more distinct layers as shown in the right of Figure 4.
>
> **W2**
>
> > The recurrent-depth design is a major practical drawback. As confirmed by the authors' runtime analysis in Table 15, the 1-layer, 12-iteration HYPER-SET model is significantly slower than a standard Transformer, despite having fewer parameters. This high latency severely limits its practicality.
>
>
> We acknowledge this extra computational cost as one of our limitation in Appendix H.3, largely due to the modulation network that learns the step sizes (~2.33ms) which is adopted only for performance reasons rather than being a theoretically necessary component in our framework. It comes inevitably with some expenses to match Transformer with "white-box" design: more parameters (CRATE [1]) or more runtime cost (ours). In fact, we have provided ablations that remove this extra network and fix the step sizes, which gives similar runtime (5.70ms in Table 17), but the performance decreases greatly (81.45 on CIFAR-10 and 51.29 on CIFAR-100 in Table 4). So, we believe there is still room for improvement in optimizing the step sizes schedule and further enhancing this minimalist implementation in future work.
>
>
> Besides Figure 4, we also provide the comparison in a **12-layer non-recurrent setting** (L12R1) below to show the practicality of our model and update the paper with Table 2, where we report results on CIFAR-10/100 with fine-tuning or training from scratch and results on large-scale ImageNet-1K. The pretraining recipe on ImageNet-1K is the same as experiments Table 1 while fine-tuning is done by adjusting batch size to 256, epochs to 50, learning rate to 1e-4, and weight decay to 1e-5.
>
> Our model can actually scale almost equally well as standard Transformer while being more parameter-efficient. This suggests our model can have the benefit of scaling in Transformer when extended practically, meanwhile being more useful in resource-constrained settings.
>
> $*$ means first pretraining on ImageNet-1K (IN-1K) and then fine-tuning on CIFAR10/100.
>
> |Model |Config (width $d$)|#Param|Acc. (IN-1K)| Acc. (CIFAR10)|Acc. (CIFAR100)| Acc. (CIFAR10)*|Acc. (CIFAR100)*|
> |:--- | :---: | :---: | :---: | :---: |:---: |:---: |:---: |
> |Transformer|L12R1 (384)|21.86M|**67.90**|87.44|62.84|**96.95**|**83.10**|
> |CRATE|L12R1 (512)|10.28M|60.96|*88.76*|63.94|94.18|77.39|
> |Ours |L12R1 (512)|8.17M |66.26|**88.82**|**64.98**|95.76|80.89|
> |Ours |L12R1 (768)|17.56M |*67.20*|88.53|*64.16*|*96.47*|*82.60*|

---

> ### Author Response · Authors · 2025-11-20
> **Response 2**
>
> **Q2**
>
> > Does the training memory cost scale linearly with the number of iterations to store activations for backpropagation?
>
> The following table shows how GPU memory on Nvidia A100 relates to forward iterations using the training recipe on IN-1K in the paper. It does not scale linearly in the smaller iterations, which we presume the modulation network and some fixed buffers predominantly take a constant amount of memory. But it roughly scales linearly when the iterations are larger than 6.
>
> |Iter.|1|2|3|4|5|6|7|8|9|10|11|12|
> |:-|:-|:-|:-|:-|:-|:-|:-|:-|:-|:-|:-|:-|
> |Mem.(MB)|13968|16036|16038|16038|16040|16042|16044|17982|20020|22058|24096|26132|
>
> **Q3**
>
> > The paper's most significant performance gap appears on the Sudoku task, where both CRATE and Energy Transformer completely fail on this task. What is the authors' hypothesis for why these other "white-box" models fail catastrophically on this reasoning task, while HYPER-SET succeeds?
>
> We first want to point out that Sudoku may be hard for some architecture to achieve non-zero results, because the 9x9 Sudoku grid has to be filled in with some exact digits to meet the constraints which may be hard to learn without proper inductive bias. As shown in Figure 1 of [2], even some foundation models has zero accuracy.
>
> As we state in Appendix H.2 about the broader impact of our paper, previous works diverge from their theoretical formulations and the design objective are inspired by domain knowledge. These "white-box" models inevitably infuse some prior assumptions or biases into the architecture but their formulation may not characterize the information processing in the right way, and the optimization-induced layer need to be implemented extremely carefully that would otherwise make the whole model more brittle than Transformer that has fewer biases.
>
> Therefore, a reasonable explanation would be that Hyper-SET succeeds because we have a better modeling or assumption about the design objective, which is based on as fundamental as maximum likelihood estimation and thus more realistic, and we rigorously follow our formulation without deviations in implementation, making the architecture better aligned with the optimization procedure. This could allow Hyper-SET to enjoy both the principled design and the expressivity of Transformer.
>
> ---
> [1] Yu et al. White-box transformers via sparse rate reduction. NeurIPS, 2023.
>
> [2] Wang et al, Hierarchical Reasoning Model, arXiv, 2025

---

> > ### Comment · Reviewer_jreb · 2025-11-27
> >
> > I thank the authors for their detailed responses. I find that the responses successfully clarify my concerns.
> >
> > Although HYPER-SET still has some disadvantages compared to Transformers, I consider this work a meaningful advancement for 'white-box' architectural design.
> >
> > I suggest that the authors include memory usage in the experiments, rather than just the number of parameters, to better support the case for resource-constrained settings.
> >
> > While I understand that the argument for HYPER-SET (recurrent) being suitable for resource-constrained settings does not contradict the claim that HYPER-SET (non-recurrent) scales well (Line 430), I suggest refining the writing in this section to ensure this distinction is clear to readers.
> >
> > I appreciate the improvements and clarifications, and I have updated my verdict accordingly.

---

> > > ### Author Response · Authors · 2025-11-28
> > >
> > > We are happy to see the reviewer's concerns have been addressed and grateful for further raising the score.
> > >
> > > We have updated the paper with the suggested clarifications from lines 422 to 430 and additional details on memory usage in Appendix D.6.

---

### Official Review · Reviewer_PVNu · 2025-11-04

**Soundness:** 2
**Presentation:** 3
**Contribution:** 2
**Rating:** 2
**Confidence:** 4

**Summary:**

This paper proposes a new transformer component, based on the principle of optimizing a hyperspherical energy.

**Strengths:**

- The paper seeks to formulate model architecture design through the principle of energy minimization, which is an interesting and ambitious attempt.
- The experiments span several domains, showing the effectiveness of the proposed method.
- This paper is well presented and easy to follow.

**Weaknesses:**

- It is unclear if the proposed "energy" is actually minimized. I don't see any empirical verification or any theoretical proof regarding this.
- Also, there is no justification for whether the energy decrease actually correlates with performance.
- The resulting architecture appears extremely similar to a standard Transformer with RMSNorm or other normalization layers. It is unclear what the "energy" formulation contributes beyond rephrasing existing operations in geometric language.

**Questions:**

N/A

---

> ### Author Response · Authors · 2025-11-20
> **Response 1**
>
> Thank you for acknowledging our attempt towards a principled model design. We will address the reviewer’s concerns and further underscore the significance of our work. **Please note we have updated the manuscript in blue color** and will refer to the new versions of tables and figures in the responses.
>
> **Weaknesses**
>
> > It is unclear if the proposed "energy" is actually minimized. I don't see any empirical verification or any theoretical proof regarding this.
>
> **Empirically**, as shown in Section 5.4 and Figure 5, we demonstrate the behavior of Attention Energy ($E_{ATTN}$) and Feedforward Energy ($E_{FF}$) in the forward pass, both separately and as a whole. **The (almost) monotonically decreasing behavior of energy suggests that the layer aligns well with the optimization objective.**
>
> **Theoretically**, we use alternating minimization (Algo. 1) to optimize the energy function ($E_{ATTN}+E_{FF}$), following CRATE [1], as mentioned in line 250.
>
> |Algo. 1||
> |:--- | :---: |
> |For a combined objective $h(y)=f(y)+g(y)$, at $t$-th iteration, execute|$u^{(t)} = y^{(t)}-\alpha_1 \nabla f(y^{(t)})$; $y^{(t+1)} = u^{(t)}-\alpha_2 \nabla g(u^{(t)})$|
>
> Crucially, this optimization has been proven in Section 4 of [2] to guarantee the decrease of the objective $h$. Since this is a general optimization procedure, it applies to our setting as well.
>
> Due to character limits, we defer the theoretical proof to the second response.
>
> > Also, there is no justification for whether the energy decrease actually correlates with performance.
>
> We want to clarify the relation between “energy” decrease and performance and justify our framework from the following perspectives.
>
> - First, existing studies [3] have shown that neural networks induced from unrolling a _task-specific_ objective—a form of energy minimization—can improve performance on that task. However, as we argue from lines 54 to 59, this approach relies on fixed task priors and lacks generality.
>
> - Second, as a response, we opt for a more general framework of energy-based learning and attempt to model the forward pass, i.e., learning good representations, as an energy-minimization process. **Learning good representations has been well-accepted for good downstream performance.** Therefore, this core challenge boils down to how to define a measure for “good” representations and, subsequently, how to quantify or explicitly compute that measure.
>
>   To this end, we formulate the objective of the forward pass _as fundamental as a joint maximum likelihood estimation_ (MLE). This MLE objective is justified in Appendix A, which results in Eq. (1) mathematically similar to evidence lower bound (ELBO) that is widely used in generative modeling. Meanwhile, in the spirit of aligning representation learning and network structure, we achieve designing Transformers via unrolling the instantiated energy functions that quantify Eq. (1). **To our knowledge, no prior works have noticed this delicate relation between this fundamental probabilistic interpretation and Transformers layer and successfully instantiated optimizable (energy) functions for designing Transformers based on this principle.**
>
> - Finally, we want to point out that it is non-trivial, if possible, to rigorously establish this correlation **quantitatively** because it is hard to isolate the effect of engineering efforts, like hyperparameter tuning, from the “goodness” of representations in terms of performance measurement, which may obfuscate the conclusion. Also, how this “goodness” objective (energy function) is optimized—whether by gradient descent like ours, or with a momentum term, or Adam–could affect performance. Therefore, as a workaround, we provide a **qualitative** justification by **showing that decreasing energy indeed corresponds to learning better representations (increase in accuracy) on the right of Figure 5**.
>
>
>
> ---
> [1] Yu et al. White-box transformers via sparse rate reduction. NeurIPS, 2023.
>
> [2] Yang et al. Transformers from an Optimization Perspective, NeurIPS 2022.
>
> [3] Shlezinger et al. Model-based deep learning. Proceedings of the IEEE, 2023

---

> ### Author Response · Authors · 2025-11-20
> **Response 2**
>
> Given the optimization of Algorithm 1 we adopt in the paper, we first restate the conclusion from [2]:
>
> > Remark 4.4. Our findings above can be summarized as follows: For sufficiently small values of $\alpha_1$, $\alpha_2$, Algorithm 1 reduces the combined objective $h$, at least provided that $y$ is a certain distance away from the optimal point $y^*$.
>
> **Proof**: We give a brief proof of the descent behavior of Algorithm 1 below.
>
> **Assumptions**:
>
> For continuous and differentiable functions $f, g:\mathbb{R}^d\rightarrow \mathbb{R}$, assume they are $L$-smooth with Lipschitz constant $L_f$ and $L_g$:
>
> $$f(y)\leq f(x)+\nabla f(x)^T(y-x)+\frac{L_f}{2}\|y-x\|^2,$$
>
> $$g(y)\leq g(x)+\nabla g(x)^T (y-x)+\frac{L_g}{2}\|y-x\|^2.$$
>
> **Bounding $h(y^{t+1})$**:
>
> For the update in Algorithm 1, we have:
> $$y^{t+1}=y^t-\alpha_1\nabla f(y^t)-\alpha_2\nabla g(u^t)=y^t-\Delta_t,$$
>
> where $\Delta_t=\alpha_1\nabla f(y^t)+\alpha_2\nabla g(u^t)$. By $L$-smoothness of $f$ and $g$, we have:
>
> $$f(y^{t+1}) \leq f(y^t)-\alpha_1\|\nabla f(y^t)\|^2-\alpha_2\nabla f(y^t)^T\nabla g(u^t)+\frac{L_f}{2}\|\Delta_t\|^2, \quad (1)$$
>
> $$g(y^{t+1}) \leq g(y^t)-\alpha_1\nabla g(y^t)^T\nabla f(y^t)-\alpha_2\nabla g(y^t)^T \nabla g(u^t)+\frac{L_g}{2}\|\Delta_t\|^2. \quad (2)$$
>
>
> Summing (1) and (2) while substituing $\Delta_t$ gives:
>
> $$
> h(y^{t+1}) \leq h(y^t)+\left(\frac{(L_f +L_g)\alpha_1^2}{2}-\alpha_1 \right) \|\nabla f(y^t)\|^2+\frac{(L_f +L_g)\alpha_2^2}{2}\|\nabla g(u^t)\|^2 + \left((L_f+L_g)\alpha_1\alpha_2-\alpha_2 \right) \nabla f(y^t)^T\nabla g(u^t)-\alpha_1\nabla f(y^t)^T\nabla g(y^t)-\alpha_2\nabla g(y^t)^T\nabla g(u^t). \quad (3)
> $$
>
> **Defining Error:**
>
> Let error $e^t=\nabla g(u^t)-\nabla g(y^t)$. Given that $g$ is $L_g$-smooth, we have:
> $$\|e^t\|\leq L_g\|u^t-y^t\|=L_g\alpha_1\|\nabla f(y^t)\|.$$
>
> Plugging $\nabla g(u^t)=\nabla g(y^t)+e^t$ into (3) yields:
>
> $$
> h(y^{t+1}) \leq h(y^t)+\left(\frac{(L_f+L_g)\alpha_1^2}{2}-\alpha_1 \right) \|\nabla f(y^t)\|^2+\left(\frac{(L_f +L_g)\alpha_2^2}{2}-\alpha_2 \right)\|\nabla g(y^t)\|^2+((L_f+L_g)\alpha_1\alpha_2-\alpha_1-\alpha_2)\nabla f(y^t)^T\nabla g(y^t)+((L_f+ L_g)\alpha_1\alpha_2-\alpha_2)\nabla f(y^t)^Te^t+((L_f+L_g)\alpha_2^2-\alpha_2)\nabla g(y^t)^Te^t+\frac{L_f+L_g}{2}\alpha_2^2\|e^t\|^2. \quad (4)
> $$
>
> **Bounding Terms in (4):**
>
> Consider sufficiently small $\alpha_1,\alpha_2$, this will make the order of $\alpha_1\alpha_2$ higher than $\alpha_1$ and $\alpha_2$, thus rendering the coefficient of each cross-term is non-positive:
>
> $$
>     (L_f+L_g)\alpha_1\alpha_2-\alpha_1-\alpha_2 \leq 0, \\
>     (L_f+L_g)\alpha_1\alpha_2-\alpha_2  \leq 0, \\
>     (L_f+L_g)\alpha_2^2-\alpha_2 \leq 0, \\
> $$
>
> with Cauchy–Schwarz inequality and $\|e^t\|\leq L_g\|u^t-y^t\|=L_g\alpha_1\|\nabla f(y^t)\|$, we get:
> $$
> ((L_f+L_g)\alpha_1\alpha_2-\alpha_1-\alpha_2)\nabla f(y^t)^T\nabla g(y^t)  \leq (\alpha_1+\alpha_2-(L_f+L_g)\alpha_1\alpha_2)\|\nabla f(y^t)\|\|\nabla g(y^t)\|, \\
> ((L_f+L_g)\alpha_2^2-\alpha_2)\nabla g(y^t)^Te^t  \leq(\alpha_2-(L_f + L_g)\alpha_2^2 )L_g\alpha_1\|\nabla f(y^t)\|\|\nabla g(y^t)\|,\\
> ((L_f+L_g)\alpha_1\alpha_2-\alpha_2)\nabla f(y^t)^Te^t  \leq(\alpha_2-(L_f+L_g)\alpha_1 \alpha_2)L_g\alpha_1\|\nabla f(y^t)\|^2,\\
> \frac{L_f+L_g}{2}\alpha_2^2\|e^t\|^2  \leq\frac{L_f+L_g}{2}\alpha_2^2L_g^2\alpha_1^2\|\nabla f(y^t)\|^2
> $$
>
> **Descent Conditions:**
>
> Substitute them to bound corresponding terms in (4), ignore the high-order term and let the coefficient be non-positive, we have the conditions for guaranteed descent on $h=f+g$:
>
> $$(L_f+L_g)\alpha_1+2L_g\alpha_2\leq2,\\
> 1/\alpha_1+1/\alpha_2\geq L_f,\\
> \alpha_1\leq1/(L_f+L_g),\\
> \alpha_2\leq1/(L_f+L_g),$$
>
> which can be satisfied if $\alpha_1,\alpha_2$ are sufficiently small.
>
> ---
>
> [2] Yang et al. Transformers from an Optimization Perspective, NeurIPS 2022.

---

> ### Author Response · Authors · 2025-11-20
> **Response 3**
>
> > The resulting architecture appears extremely similar to a standard Transformer with RMSNorm or other normalization layers. It is unclear what the "energy" formulation contributes beyond rephrasing existing operations in geometric language.
>
> - First, we want to highlight that our instantiated architecture in Figure 2, differs from standard Transformers in several ways: 1) our bidirectional $\operatorname{softmax}(\cdot)$ for self-attention-both row-wise and column-wise vs. Transformer column-wise only; 2) our weight-tying and sharing $W_Q=W_K=W_V=W_O^T$ in attention, and the weight sharing in feedforward $W_{\text{FF1}} = W_{\text{FF2}}^T$; 3) our learned step sizes. These structural biases together are induced from and directly contribute to minimizing the defined energy functions.
>
> - Second, our energy-based perspective opens up new avenues for innovation in model design. By defining alternative energy functions to quantify Eq.(1), we can design novel components beyond existing operations, such as **linear attention** and **gated feedforward** as mentioned in Section 5.5 and Appendix G.1. This is a significant advantage over traditional heuristic-based designs, as it allows for broader energy-based designs—such as kernel-based alternatives to $\operatorname{logsumexp}(\cdot)$ in Eq.(3)—enabling principled generalizations beyond standard attention mechanisms, thus showcase promising potential of our formulation.

---

> ### Author Response · Authors · 2025-11-28
> **Kindly Request for Follow-Up on Rebuttal**
>
> Dear Reviewer PVNu,
>
> We kindly invite you to take a look at our rebuttal when convenient. If our response addresses your concerns, we would greatly appreciate any further feedback and a possible score update. Thank you for your time.
>
> Authors

---

### Author Response · Authors · 2025-12-02
**Final Comments to AC**

Dear AC,

We sincerely appreciate your time in reviewing our rebuttal and follow-up discussion. We summarize how we addressed the key concerns raised by the reviewers below.

Our review scores are:
- Reviewer PVNu: 2
- Reviewer jreb: 4 (-> 6)
- Reviewer Dhku: 8
- Reviewer fUR7: 4 (-> 6)

**Reviewer jreb** and **fUR7** primarily raised concerns about the practicality of Hyper-SET (parameter efficiency, latency, memory), requested explanations on Sudoku performance, and sought more pragmatic results in non-recurrent-depth settings. We addressed their concerns by justifying its value under resource-constrained settings with detailed profiling, articulating reasonable explanations on why we succeed on Sudoku, and providing results on ImageNet-1K in a non-recurrent setting. **Both of them have raised their score from 4 to 6** (reverted back due to system issues), as evidenced by their comments:
- "...I appreciate the improvements and clarifications, and *I have updated my verdict accordingly*." (Reviewer jreb)
- "...because of this improvement and because many of my questions/weaknesses were addressed, *I will increase the score*." (Reviewer fUR7)


**Reviewer PVNu** raised concerns on the theoretical justifications and empirical verifications regarding energy minimization, its correlation to downstream performance, and whether the method extends beyond existing operations. We **pointed to the empirical verifications already in Figure 5** of the original manuscript that energy indeed decreases, **provided theoretical proof** on the descent behaviour of the unrolling optimization we adopted and **qualitative evidence for the positive correlation to performance**, and clarified the **extensibility to design novel components like linear attention and gated feedforward**. While we believe these responses effectively addressed this reviewer's concerns, **we have not yet received any feedback** during our rebuttal. We hope you can take this into consideration when making the final decision.

**Reviewer Dhku** suggested more discussion on the outlook in practical scalability, better positioning in the literature, and the intuition behind Sudoku performance. We updated the manuscript with these discussions and an additional related work subsection on depth-recurrence in Transformers.

---
Accordingly, we have made the following changes with blue color in the revised manuscript:
- Added correlation curves between "energy" decrease and downstream performance in the right panel of Figure 5. (Reviewer PVNu)
- Added Table 2 for comparison in a more practical non-recurrent setting with discussion in Section 5.2. (Reviewer jreb, fUR7)
- Added more insights and discussion regarding the superiority over baseline models on Sudoku in Section 5.1. (Reviewer jreb, Dhku, fUR7)
- Added a subsection on depth-recurrence in Transformers in related work. (Reviewer Dhku)
- Updated runtime analysis in Appendix D.6 with memory cost and discussion on the trade-off. (Reviewer jreb)

---
**To further highlight, no prior works, to our knowledge, have noticed this delicate relation between this fundamental probabilistic interpretation and Transformers layer and successfully instantiated optimizable (energy) functions for designing Transformers based on this principle.** We hope our work contributes significantly to the field of principled and interpretable architectural design. Thank you again for your time and thoughtful engagement.

Best,

Authors

---

> ### Author Response · Authors · 2025-12-03
> **Clarification on Score Changes and Conduct Compliance**
>
> We’d like to finally bring attention to AC that **Reviewer jreb and fUR7 raised their score before the identities leakage issue broke out**. Our rebuttal and discussion strictly followed the ICLR code of conduct.
>
> Best,
>
> Authors

---

### Meta-Review · Area_Chair_3N4g · 2026-01-06

**Summary:**

Initial reviews were mixed (2,4,4,8).

Reviewer PVNu questions whether the energy used for the derivation is actually minimized by the algorithm and whether any architectural insights are gained from the work over standard transformer architectures.

Reviewer jreb raises concerns about empirical performance and computational cost but notes a strong contribution to the white-box design literature.

Reviewer Dhku is largely positive in their review and notes potential issues with scaling the model to large scale datasets.

Reviewer fUR7 also notes concerns about scaling to large datasets and the potential limitation of a single-layer recurrent architecture as well as issues with whether the baseline comparisons are sufficiently tuned.

**Reviewer Concerns:**

The authors respond to concerns about what is being minimized via the energy unrolling architecture by noting prior work has shown that this converges provided enough iterations (or in this case recursive calls) are performed.  Moreover, they note that their energy formulation differs from standard transformers via weight sharing and by applying softmax on both the rows and columns of the attention matrix, along with the general advantages of white-box design that allow for more principled exploration of design choice and the resulting architectures.

In terms of empirical performance and scaling to large datasets the authors have provided additional experiments showing competitive performance on ImageNet-1K and in non-recursive settings.

In my view, a majority of the more substantial criticisms raised by the reviewers have been addressed.

**Reviewer Scores:**

Two of the borderline reviewers note increasing their scores, with the most negative reviewer (PVNu) not responding before the discussion was closed.

Overall I believe the authors have responded well to the bulk to the critiques raised by the reviewers, and many of the criticisms raised by PVNu specifically appear to be largely addressed.  As a result, I am fairly confident that during a discussion period a consensus would be reached to accept the paper.

---

### Decision · Program_Chairs · 2026-01-26

Accept (Poster)